# PROVABLE DEFENSE AGAINST GEOMETRIC TRANSFORMATIONS

**Rem Yang[1], Jacob Laurel[1], Sasa Misailovic[1], Gagandeep Singh[1,2]**
[1]University of Illinois Urbana-Champaign, [2]VMware Research
{remyang2,jlaurel2,misailo,ggnds}@illinois.edu

## ABSTRACT

Geometric image transformations that arise in the real world, such as scaling and rotation, have been shown to easily deceive deep neural networks (DNNs). Hence, training DNNs to be certifiably robust to these perturbations is critical. However, no prior work has been able to incorporate the objective of deterministic certified robustness against geometric transformations into the training procedure, as existing verifiers are exceedingly slow. To address these challenges, we propose the first provable defense for deterministic certified geometric robustness. Our framework leverages a novel GPU-optimized verifier that can certify images between $60\times$ to $42,600\times$ faster than existing geometric robustness verifiers, and thus unlike existing works, is fast enough for use in training. Across multiple datasets, our results show that networks trained via our framework consistently achieve state-of-the-art deterministic certified geometric robustness and clean accuracy. Furthermore, for the first time, we verify the geometric robustness of a neural network for the challenging, real-world setting of autonomous driving.

## 1 INTRODUCTION

Despite the widespread success of deep neural networks (DNNs), they remain surprisingly susceptible to misclassification when small adversarial changes are applied to correctly classified inputs (Goodfellow et al., 2015; Kurakin et al., 2018). This phenomenon is especially concerning as DNNs are increasingly being deployed in many safety-critical domains, such as autonomous driving (Bojarski et al., 2016; Sitawarin et al., 2018) and medical imaging (Finlayson et al., 2019).

As a result, there have been widespread efforts aimed at formally verifying the robustness of DNNs against norm-based adversarial perturbations (Gehr et al., 2018; Singh et al., 2019; Weng et al., 2018; Zhang et al., 2018) and designing novel mechanisms for incorporating feedback from the verifier to train provably robust networks with deterministic guarantees (Gowal et al., 2019; Mirman et al., 2018; Xu et al., 2020; Zhang et al., 2020). However, recent works (Dreossi et al., 2018; Engstrom et al., 2019; Hendrycks & Dietterich, 2019; Kanbak et al., 2018; Liu et al., 2019) have shown that geometric transformations – which capture real-world artifacts like scaling and changes in contrast – can also easily deceive DNNs. No prior work has formulated the construction of a deterministic provable defense needed to ensure DNN safety against geometric transformations. Further, existing deterministic verifiers for geometric perturbations (Balunovic et al., 2019; Mohapatra et al., 2020) are severely limited by their scalability and cannot be used during training for building provable defenses. Probabilistic geometric robustness verifiers (Fischer et al., 2020; Hao et al., 2022; Li et al., 2021) are more scalable but may be inadequate for safety-critical applications like autonomous driving, since they may falsely label an adversarial region as robust. These limitations have prevented the development of deterministic provable defenses against geometric transformations thus far.

**Challenges.** Training networks to be certifiably robust against geometric transformations carries multiple challenges that do not arise with norm-based perturbations. First, geometric transformations are much more difficult to formally reason about than $\ell_p$ perturbations, as unlike an $\ell_p$-ball, the adversarial region of a geometric transformation is highly nonuniform and cannot be directly represented as a symbolic formula encoding a convex shape. Computing this adversarial input region for geometric perturbations is indeed the main computational bottleneck faced by existing geometric robustness verifiers (Balunovic et al., 2019; Mohapatra et al., 2020), thus making the overall

verification too expensive for use during training. Hence, training DNNs for deterministic certified robustness against geometric perturbations requires not only formulating the construction of a provable defense, but also completely redesigning geometric robustness verifiers for scalability.

**This Work.** To address the outlined challenges, we propose Certified Geometric Training (CGT), a framework for training neural networks that are deterministically certified robust to geometric transformations. The framework consists of (1) the Fast Geometric Verifier (FGV), a novel method to perform geometric robustness certification that is orders of magnitude faster than the state-of-the-art and (2) computationally efficient loss functions that embed FGV into the training procedure.

We empirically evaluate our method on the MNIST (LeCun et al., 1998), CIFAR10 (Krizhevsky, 2009), Tiny ImageNet (Le & Yang, 2015), and Udacity self-driving car (Udacity, 2016) datasets to demonstrate CGT's effectiveness. Our results show that CGT-trained networks consistently achieve state-of-the-art clean accuracy and certified robustness; furthermore, FGV is between $60\times$ to $42,600\times$ faster than the state-of-the-art verifier for certifying each image. We also achieve several breakthroughs: (1) FGV enables us to certify deterministic robustness against geometric transformations on entire test sets of 10,000 images, which is more than $50\times$ the number of images over existing works (100 in Balunovic et al. (2019) and 200 in Mohapatra et al. (2020)); (2) we are the first to scale deterministic geometric verification beyond CIFAR10; and (3) we are the first to verify a neural network for autonomous driving under realistic geometric perturbations. Our code is publicly available at `https://github.com/uiuc-arc/CGT`.

## 2 RELATED WORK

**Geometric Robustness Certification.** Certification of geometric perturbations (such as image rotations or scaling) going beyond $\ell_p$-norm attacks have recently begun to be studied in the literature. There are two main approaches to formally verifying the geometric robustness of neural networks: deterministic and probabilistic. Most recent works on geometric robustness verification use randomized smoothing-based techniques to obtain *probabilistic* guarantees of robustness (Fischer et al., 2020; Hao et al., 2022; Li et al., 2021). While these approaches can scale to larger datasets, their analyses are inherently *unsound*, i.e., they may falsely label an adversarial region as robust. For safety-critical domains, this uncertainty may be undesirable. Furthermore, such guarantees are obtained over a smoothed version of a base network, which at inference time requires sampling (i.e., repeatedly evaluating) the network up to 10,000 times per image, thereby introducing significant runtime or memory overhead (Fischer et al., 2020). Our work focuses on *deterministic* verification, whose analysis is always guaranteed to be correct and whose certified network incurs no overhead during inference. In particular, this type of certificate always holds against any adaptive attacker. However, the existing deterministic geometric robustness verifiers (Balunovic et al., 2019; Mohapatra et al., 2020) are exceedingly slow and thus unsuitable for training, due to the high cost of abstracting the geometric input region (even before propagating it through the network). Singh et al. (2019) study deterministic robustness against rotations, but their results are subsumed by Balunovic et al. (2019). While some works have focused on scaling verifiers (Müller et al., 2021; Xu et al., 2020), they have not targeted geometric robustness as we do.

**Provable Defenses Against Norm-based Perturbations.** Prior works (Gowal et al., 2019; Mirman et al., 2018; Xu et al., 2020; Zhang et al., 2020) incorporate verification explicitly into the training loop so that the trained networks become easier to verify. Typically, this task is accomplished by formulating a loss function that strikes a balance between the desired formal guarantees (i.e., high certified robustness) and clean accuracy (i.e., the network's accuracy on the original dataset). To ensure quick loss computation over a large set of training images, interval bound propagation (IBP) (Gowal et al., 2019; Mirman et al., 2018) is the most popular technique. However, previous IBP works mainly consider norm-based perturbations, which do not capture the highly nonuniform adversarial input regions that characterize geometric transformations.

## 3 BACKGROUND

We denote a neural network as a function $f\colon \mathbb{R}^{C\times H\times W} \to \mathbb{R}^{n_o}$ from an $H \times W$ image with $C$ channels to $n_o$ real values. We focus on feedforward networks, but our approach is general and equally applicable to other architectures. Let $x \in \mathbb{R}^{C\times H\times W}$ denote an input image and $y$ denote its

corresponding label. For the remainder of the paper, we write interval variables with a tilde: e.g., $\tilde{x} = [\underline{x}, \overline{x}] = \{x : \underline{x} \le x \le \overline{x}\}$ is an interval with $\underline{x}$ and $\overline{x}$ as lower and upper bounds, respectively. First, we detail the geometric transformations that our work considers.

## 3.1 GEOMETRIC TRANSFORMATIONS

A geometric image perturbation is a function $P \colon \mathbb{R}^{C \times H \times W} \times \mathbb{R}^{|\theta|} \to \mathbb{R}^{C \times H \times W}$, which takes an input image $x$ and parameters $\theta$ (e.g., angle of rotation, amount of brightness), then produces the geometrically perturbed image $x'$. We consider *interpolated* transformations – rotation, translation, scaling, and shearing – and *pixelwise* transformations – contrast and brightness. For ease of expression, since these transformations (Eqs. 2 and 3 below) apply independently to each image channel, we write $x_{i,j}$ to denote $x$'s pixel in the $i^{\text{th}}$ row and $j^{\text{th}}$ column of an arbitrary channel.

**Interpolated Transformations.** Interpolated transformations involve an affine transformation on each pixel's row and column indices, followed by an interpolation operation. As these operations are performed in the 2D plane, we first interpret the row and column indices $(i, j)$ as points $(u, v) \in \mathbb{R}^2$, where the $u$-axis is the horizontal axis and the $v$-axis is the vertical axis. Here, we define functions $\phi_u(j) = j - (W - 1)/2$ and $\phi_v(i) = (H - 1)/2 - i$, which convert zero-indexed $i, j$ indices to $u, v$ coordinates with respect to the center of an $H \times W$ image. Let $T_\theta \colon \mathbb{R}^2 \to \mathbb{R}^2$ be an invertible affine transformation (e.g., rotation, translation) parameterized by $\theta$ (e.g., rotation angle, amount of horizontal shift). The equations for each affine transformation are given in Eqs. 10-13 in Appendix A. To compose multiple perturbations, we compose their respective affine transformations; for example, scaling by $\lambda$, rotating by $\varphi$, then shearing by $\gamma$ results in the transformation $T_\theta(u, v) = (T_\gamma^{\text{shear}} \circ T_\varphi^{\text{rotate}} \circ T_\lambda^{\text{scale}})(u, v)$ where $\theta = (\lambda, \varphi, \gamma)$. Having converted row-column indices to $\mathbb{R}^2$, we compute for each location $(\phi_u(j), \phi_v(i))$ the (real-valued) coordinate that maps to this location under $T_\theta$; we can obtain this coordinate as $(u', v') = T_\theta^{-1}(\phi_u(j), \phi_v(i))$, where $T_\theta^{-1}$ is the inverse transformation. Since these transformed coordinates may not align exactly with integer-valued pixel indices, we must interpolate. We consider the bilinear interpolation kernel of Jaderberg et al. (2015), given as:

$$I_x(u, v) = \sum_{n=0}^{H-1} \sum_{m=0}^{W-1} x_{n,m} \cdot \max(0, 1 - |v - \phi_v(n)|) \cdot \max(0, 1 - |u - \phi_u(m)|) \qquad (1)$$

The value of each pixel in the interpolated image $x'$ is then:

$$x'_{i,j} = I_x\big(T_\theta^{-1}(\phi_u(j), \phi_v(i))\big) \qquad (2)$$

**Pixelwise Transformations.** The cumulative effects of contrast and brightness acting on the pixel $x_{i,j}$ are given by the respective contrast and brightness perturbation parameters $\alpha, \beta \in \mathbb{R}$, as described in Balunovic et al. (2019); Mohapatra et al. (2020):

$$x'_{i,j} = \min\big(1, \max\big(0, (1 + \alpha) \cdot x_{i,j} + \beta\big)\big) \qquad (3)$$

## 3.2 INTERVAL BOUND PROPAGATION

For each pixel in the input and each neuron in the DNN, verification with interval bound propagation (IBP) (Gowal et al., 2019) associates an interval bounding its minimum and maximum values. A sound verifier propagates intervals through the network from the input to the output layer by evaluating the network's layers using interval arithmetic; we detail these operations in Appendix B.

## 3.3 CERTIFYING GEOMETRIC ROBUSTNESS USING IBP

We now show how to certify the geometric robustness of classification and regression networks.

**Certified Classification Robustness Against Geometric Transformations.** Given an interval $\tilde{x}$ enclosing the set of possible perturbations on the input and a classifier $f$, we denote the worst-case output vector as $\hat{f}(\tilde{x})$, where (interpreting all operations via interval arithmetic) the correct class's entry is the lower bound and all other entries are the upper bounds of the network's output:

$$\hat{f}_y(\tilde{x}) = \underline{f_y}(\tilde{x}) \quad \text{and} \quad \hat{f}_j(\tilde{x}) = \overline{f_j}(\tilde{x}) \ \ \forall j \in \{1, 2, \ldots, n_o\} \setminus \{y\} \qquad (4)$$

We say that $f$ is certifiably robust for $\tilde{x}$ if, even in the worst case, we can still guarantee that the network classification is correct: $y = \arg\max_i \hat{f}_i(\tilde{x})$. To certify geometric robustness on an image $x$ requires computing its bounds as $\tilde{x} = P(x, \tilde{\theta})$, where $\tilde{\theta} = [\underline{\theta}, \overline{\theta}]$ is an interval vector bounding the range of geometric perturbation parameters, and the geometric perturbation $P$ is interpreted over interval arithmetic. However, for a given range of perturbation parameters $\tilde{\theta}$ for which we wish to verify robustness, the interval width $\overline{\theta} - \underline{\theta}$ may be too large for the direct evaluation of $\hat{f}(P(x, \tilde{\theta}))$ to yield bounds that are precise enough for successful certification. Hence, like other geometric robustness verifiers (Balunovic et al., 2019; Mohapatra et al., 2020), we employ parameter splitting.

We subdivide the entire range of parameters $\tilde{\theta}$ into $K$ smaller disjoint intervals $\{\tilde{\theta}_1, \tilde{\theta}_2, \ldots, \tilde{\theta}_K\}$ where $\tilde{\theta} = \bigcup_{k=1}^{K} \tilde{\theta}_k$, and certify each split $\tilde{\theta}_k$ independently. If verifying all splits succeeds, then certification holds on the entire parameter range $\tilde{\theta}$. As interval bounds are constants and do not symbolically depend on the input parameters (unlike DeepG (Balunovic et al., 2019)), propagating these bounds is highly efficient and can be done on a large number of parameter splits for improved precision. Accounting for splitting, we say that $f$ is certifiably robust for $P(x, \tilde{\theta})$ if:

$$y = \arg\max_i \hat{f}_i(P(x, \tilde{\theta}_k)) \ \ \forall k \in \{1, 2, \ldots, K\} \tag{5}$$

**Certified Regression Bound Against Geometric Transformations.** Distinct from classifiers, as regression tasks do not have a strict notion of correctness, our goal is to verify whether a network's outputs are within a range close to the ground truth. Hence, our certification problem is to directly bound the network outputs. For a regression network $f$ (interpreted over interval arithmetic), the certified output range over the input $P(x, \tilde{\theta})$ is the smallest interval containing the union of all splits' output bounds:

$$\bigcup_{k=1}^{K} \left\{ f(P(x, \tilde{\theta}_k)) \right\} \tag{6}$$

## 4 CERTIFIED GEOMETRIC TRAINING

We now describe the formulation of our provable defense and fast geometric verifier (FGV) that comprise our training framework for certified geometric robustness.

### 4.1 ROBUST LOSS FOR CLASSIFICATION AND REGRESSION NETWORKS

**Training Classification DNNs.** The key to incorporating geometric robustness guarantees into training lies in formulating certification as part of the loss function. Since $\hat{f}$ represents the worst-case classification output under a range of perturbed inputs, we can use it in the training loss to guide the parameter updates toward obtaining provably robust DNNs. To account for parameter splitting during certification (which is unique to our geometric setting), we formulate our training loss to enforce *local* robustness at the level of individual input splits. To certify the network across the entire desired range $\tilde{\theta} = [\underline{\theta}, \overline{\theta}]$, we enforce this local property across all splits. Furthermore, similar to other works on certified training, we also need to enforce high clean accuracy. This yields the following formulation for the *ideal* robust classification loss:

$$L_{ci}(x, y) = \kappa \cdot \ell\big(f(x), y\big) + \left(\frac{1-\kappa}{K}\right) \cdot \sum_{k=1}^{K} \ell\big(\hat{f}(P(x, \tilde{\theta}_k)), y\big) \tag{7}$$

where $\ell$ can be any classification loss function (we use the cross-entropy loss) and $\kappa \in [0, 1]$ governs the relative weighting between the clean accuracy and geometric robustness terms (with higher $\kappa$ prioritizing clean accuracy).

In practice, the loss in Eq. 7 is too computationally expensive, since the runtime scales linearly with the number of splits, which often needs to be large to ensure precise certification. As a remedy, we instead enforce the robustness property *stochastically* using data augmentation in conjunction with a randomized sampling of interval splits. We uniformly sample a scalar perturbation amount $\theta \sim \mathcal{U}(\underline{\theta}, \overline{\theta})$ and compute a local interval split $\tilde{\theta}_l = [\theta - \nu, \theta + \nu]$, where $\nu$ is a hyperparameter

vector governing the interval size of each perturbation parameter. We then compute the *tractable* robust classification loss as:

$$L_{ct}(x,y) = \kappa \cdot \ell\big(f(P(x,\theta)), y\big) + (1-\kappa) \cdot \ell\big(\hat{f}(P(x,\tilde{\theta}_l)), y\big) \tag{8}$$

Since we sample a different $\theta$ for each mini-batch of training samples, this approach will, on average, effectively enforce local robustness over the entire parameter range (hence leading to global robustness). As in prior works (Gowal et al., 2019; Xu et al., 2020; Zhang et al., 2020), we can vary $\kappa$ with the training iteration, temporally changing the weighting between clean accuracy and robustness. The hyperparameter vector $\nu$ is akin to $\epsilon$ in the $\ell_\infty$-norm case, but it governs the size of a geometric ball $P(x, \tilde{\theta}_l)$ rather than an $\epsilon$-ball; note that $\nu$ is a vector, since the interval size corresponding to each geometric transformation may be different. We show that this hyperparameter is easy to determine. Finally, while this loss function incorporates only IBP to propagate the geometric region through the network, it can be easily adapted to other provable training methods like CROWN-IBP (Zhang et al., 2020) by substituting the $\epsilon$-balls in their loss functions with our formulation of local geometric balls.

**Training Regression DNNs.** Since our goal is to minimize both the width of the certified output bound as well as its distance to the ground truth, the ideal scenario would be when the certified lower and upper bounds coincide at the ground truth. Hence, we can essentially treat the certified bounds as worst-case network outputs and minimize both the lower and upper bounds' distances to the label. With similar insights gleaned from the classification loss function, we thus formulate the robust regression loss as:

$$L_r(x,y) = \kappa \cdot \ell\big(f(P(x,\theta)), y\big) + (1-\kappa) \cdot \frac{\ell\big(\underline{f}(P(x,\tilde{\theta}_l)), y\big) + \ell\big(\overline{f}(P(x,\tilde{\theta}_l)), y\big)}{2} \tag{9}$$

where $\ell$ can be any regression loss function (we use the mean squared error).

While we experimentally focus on the more common interpolated transformations, our loss formulations can also be used for other parameterized semantic perturbations on which interval over-approximations for each pixel can be computed, such as Gaussian blur or color space perturbations.

## 4.2 Fast Geometric Verifier

A key technical challenge arises during the computation of the loss functions in Eqs. 8 and 9: we need to perform geometric robustness certification on images at every iteration of training. Certification against geometric transformations requires two key steps: (1) obtaining bounds on the set of perturbed images obtainable after applying geometric perturbation $P$ and (2) propagating these bounds through a neural network. The key bottleneck of existing geometric verifiers, which renders them unusable for training, lies in the first step (which we show in Section 5.3). This bottleneck stems from the fact that when computing the interpolation, the sequence of arithmetic computations performed at each pixel can be drastically different depending on the pixel location, which has precluded existing approaches from leveraging parallelism. Thus, a core part of our contribution is designing a novel and efficient *GPU-parallelizable* method for computing interpolated transformations over interval bounds: serving the dual purpose of speeding up verification and being able to utilize these bounds during training. Algorithm 1 presents the pseudocode. Additionally, we also show a running example in Appendix C.1. By default, areas of an interpolated image with no corresponding source pixel will be padded with zero, as in Balunovic et al. (2019). However, our algorithm is agnostic to the particular padding mode: we discuss how to handle other padding strategies (e.g., replicating the border pixel values) in Appendix C.2.

We mathematically decompose the interpolated transformations (Eqs. 1 and 2) into three parts which are all GPU-parallelizable: computing the coordinates of each pixel location under inverse transformation $T_\theta^{-1}$, calculating the interpolation distances $\max(0, 1-|v-\phi_v(n)|) \cdot \max(0, 1-|u-\phi_u(m)|)$, and finally obtaining the resulting interpolated images. Our key insight is that the first two steps are solely dependent on the transformation $T_\theta$ and the height and width of the images, but *not* the image pixel values themselves. Hence, these computations need only be done once for a given transformation range, amortizing the cost for *any* number of images in a batch. We now detail each part.

**Inverse Coordinates ①.** We first determine the inverse coordinates $(\tilde{u}', \tilde{v}') = T_{\tilde{\theta}}^{-1}(\phi_u(j), \phi_v(i))$ for each $(i,j)$ pixel index, where $0 \le i < H$ and $0 \le j < W$ for images with height $H$ and

---

**Algorithm 1** Fast interval interpolated transformation.

    **Input:** $X \in [0,1]^{N \times C \times H \times W}$, a batch of $N$ images with dimension $C \times H \times W$
         $T_{\tilde{\theta}}$, an interpolated transformation with interval parameters $\tilde{\theta}$
    **Output:** $\tilde{X}' \in [[0,1]^{N \times C \times H \times W}, [0,1]^{N \times C \times H \times W}]$, a batch of transformed interval images
  1: **procedure** MAKEINTERPGRID($H, W, T_{\tilde{\theta}}$)

  ①
  2:    $(i, j) \leftarrow ([0, 1, \ldots, H-1], [0, 1, \ldots, W-1])$
  3:    $(u, v) \leftarrow (j - (W-1)/2, (H-1)/2 - i)$
  4:    $(U, V) \leftarrow ([\underbrace{u^T, u^T, \ldots, u^T}_{H \text{ times}}]^T, [\underbrace{v^T, v^T, \ldots, v^T}_{W \text{ times}}])$
  5:    $(\tilde{U}', \tilde{V}') \leftarrow T_{\tilde{\theta}}^{-1}(U, V)$

  ②
  6:    $(\tilde{U}'_r, \tilde{V}'_r) \leftarrow (\tilde{U}'.\text{reshape}(HW, 1, 1), \tilde{V}'.\text{reshape}(HW, 1, 1))$
  7:    $\tilde{G} \leftarrow \max(0, 1 - |\tilde{V}'_r - V|) \odot \max(0, 1 - |\tilde{U}'_r - U|)$

  ③
  8:    $z \leftarrow \text{count\_nonzeros}(\tilde{G}, \dim = (1, 2))$
  9:    $\tilde{g} \leftarrow \text{flatten}(\tilde{G})$
  10:    $q \leftarrow \text{get\_nonzero\_indices}(\tilde{g})$
  11:    $\tilde{w} \leftarrow \tilde{g}[q]$
  12:    $(r, c) \leftarrow (\lfloor (q \mod HW)/W \rfloor, (q \mod HW) \mod W)$
  13:    **return** $\tilde{G}_s \leftarrow (r, c, \tilde{w}, z)$
  14: **end procedure**
  15:
  16: **procedure** INTERPOLATE($X, \tilde{G}_s$)
  17:    $(r, c, \tilde{w}, z) \leftarrow \tilde{G}_s$

  ④
  18:    $\tilde{S} \leftarrow \tilde{w} \odot X[:, :, r, c]$
  19:    $\tilde{X}'_f \leftarrow \text{split\_and\_sum}(\tilde{S}, \dim = 2, \text{sizes} = z)$
  20:    **return** $\tilde{X}' \leftarrow \tilde{X}'_f.\text{reshape}(N, C, H, W)$
  21: **end procedure**

---

width $W$. The inverse transformations (shown in Appendix A Eqs. 10-13) are pixelwise, hence this step can immediately be parallelized by defining the matrices $U, V \in \mathbb{R}^{H \times W}$ where $U(i, j) = \phi_u(j)$ and $V(i, j) = \phi_v(i)$, then applying $T_{\tilde{\theta}}^{-1}$ over this whole grid of coordinates. Since $T_{\tilde{\theta}}^{-1}$ is parameterized by an interval range of parameters $\tilde{\theta}$, this inverse transformation and all subsequent arithmetic operations are interpreted via interval arithmetic.

**Interpolation Grid ② and Exploiting Sparsity ③.** For each $(\tilde{u}', \tilde{v}')$, we next calculate the terms $\max(0, 1 - |\tilde{v}' - \phi_v(n)|) \cdot \max(0, 1 - |\tilde{u}' - \phi_u(m)|)$ in Eq. 1; we call these terms *interpolation distances* and denote them $\tilde{d}_{n,m}^{\tilde{u}', \tilde{v}'}$. Here lies a key difference between our interpolation formulation and that of sequential implementations. In the sequential case, one need only compute $\tilde{d}_{n,m}^{\tilde{u}', \tilde{v}'}$ for $n$ where $\underline{v}' \leq \phi_v(n) \leq \overline{v}'$ and $m$ where $\underline{u}' \leq \phi_u(m) \leq \overline{u}'$; this is because all other values of $n, m$ will evaluate to a distance of 0. Yet since these ranges of $n, m$ can be vastly different for each $(\tilde{u}', \tilde{v}')$, the only way to parallelize the *simultaneous* computation of these distances for all inverse coordinates is to, for each $(\tilde{u}', \tilde{v}')$, compute $\tilde{d}_{n,m}^{\tilde{u}', \tilde{v}'}$ for all $0 \leq n < H, 0 \leq m < W$. We term these computed distances the *interpolation grid* $\tilde{G}$. However, since most of the entries in $\tilde{G}$ will be 0 (i.e., the degenerate interval $[0, 0]$) – typically more than 99% of them – we design a *custom sparse tensor representation* of $\tilde{G}$ so that when interpolating, only the nonzero entries will be multiplied with image pixel values (i.e., computing $x_{n,m} \cdot \tilde{d}_{n,m}^{\tilde{u}', \tilde{v}'}$ only when the distance is nonzero). First, for each nonzero $\tilde{d}_{n,m}^{\tilde{u}', \tilde{v}'}$, we must know its *location* (i.e., $n, m$) so that we can multiply its value with the corresponding image pixels in the same location. To do so, we convert $\tilde{G}$ to COO (coordinate) format, a 3-tuple of vectors: $\tilde{w}$ which stores the nonzero values of $\tilde{G}$, and $r, c$ which store the nonzero values' row and column indices $n, m$. However, once we actually interpolate across images and obtain the summands $\tilde{s}_{n,m}^{\tilde{u}', \tilde{v}'} = x_{n,m} \cdot \tilde{d}_{n,m}^{\tilde{u}', \tilde{v}'}$, we need to know which summands contribute to the same pixel (i.e., have the same $\tilde{u}'$ and $\tilde{v}'$) and should thus be added together; this is challenging since the grid values have been flattened. To solve this issue, we store an additional vector $z$ that records the number of nonzero interpolation distances for each $(\tilde{u}', \tilde{v}')$.

**Obtaining Interpolated Images ④.** After calling MAKEINTERPGRID (which only needs to be done once for a given set of transformations) and obtaining $\tilde{G}_s = (r, c, \tilde{w}, z)$, we can now efficiently interpolate across any batch of images $X$. For all batch and channel dimensions, we obtain the pixel values at locations corresponding to the interpolation distances in $\tilde{w}$ (by indexing the last two dimensions of $X$ with $r, c$) and elementwise multiply with $\tilde{w}$ to obtain $\tilde{S}$, which contains the values of all summands across all pixels. To recover the final pixel values, we sum the terms belonging to the same pixel (i.e., the $\tilde{s}_{n,m}^{\tilde{u}',\tilde{v}'}$ that have the same $\tilde{u}', \tilde{v}'$) together. To do so, we split the last dimension of $\tilde{S}$ into $HW$ chunks $\{h_i\}_{i=1}^{HW}$, where each $h_i$ has length $z_i$; then, each chunk's sum is exactly the final pixel value. Since this last dimension has been flattened to take advantage of sparsity, we reshape it back to dimension $H \times W$ to obtain the final interpolated images.

Finally, we use IBP (during both training and certification) to propagate the computed geometric bounds through the neural network.

## 5 EVALUATION

We evaluate the effectiveness of CGT over multiple datasets, network architectures, and transformations. We implemented CGT atop PyTorch (Paszke et al., 2019) and use `auto_LiRPA` (Xu et al., 2020) to propagate our computed geometric perturbation bounds through neural networks.

**Datasets and Architectures.** We evaluate our approach on the MNIST, CIFAR10, Tiny ImageNet, and Udacity self-driving car datasets. The first three are image classification tasks, while the last is a regression task that predicts a steering angle given a driving scene image. On MNIST and CIFAR10, we use the same architectures from DeepG (Balunovic et al., 2019) as to compare directly with their results. On Tiny ImageNet, we use the 7-layer convolutional network (CNN7) and WideResNet architectures from Xu et al. (2020). On Udacity, we use the classic Nvidia architecture from Bojarski et al. (2016). Details are in Appendix D.1.

**Metrics.** Our primary metrics for a classifier are (1) its clean accuracy, (2) its certified robustness under geometric transformations, and (3) the certification time on the test set. For a regression network, we utilize the analogous metrics of (1) mean absolute error (MAE), (2) certified MAE, i.e., the larger of the certified lower and upper bounds' MAEs, as well as (3) certification time. We also measure per-epoch runtime and GPU memory usage during training (shown in Appendix E.4).

**Baselines.** To the best of our knowledge, training a DNN via PGD (Madry et al., 2018) combined with data augmentation (denoted PGD/A) and certifying it with DeepG produces the current state-of-the-art deterministic geometric robustness and accuracy. However, the DeepG verifier does not scale beyond CIFAR10; hence, we compare our approach to theirs on just the MNIST and CIFAR10 datasets, using the sets of transformations from their work. Notably, DeepG only certifies 100 images (due to long runtimes), while we certify full test sets of 10,000 images. The work of Semantify-NN (Mohapatra et al., 2020) is strictly less precise than DeepG and also only handles a single interpolated transformation (rotation); thus, we only compare certification with DeepG. However, they propose an algorithm to compute the interval over-approximation of images under rotation, and we provide an ablation study comparing the speed of Algorithm 1 to theirs.

**Hyperparameters.** We show the training and certification hyperparameters in Appendices D.2 and D.3 and explain how to tune these hyperparameters in Appendix D.4.

**Hardware.** We trained and certified all networks (except WideResNet) on a machine with a 2.40GHz 24-core Intel Xeon Silver 4214R CPU with 192GB of main memory and one Nvidia A100 GPU with 40GB of memory. All baseline results were also run on the same hardware for fair comparisons. For WideResNet, we used the same CPU with four A100 GPUs.

### 5.1 MNIST AND CIFAR10

Table 1 presents the comparison of our approach with DeepG; asterisks denote DeepG certification results over a subset of 100 images (since their approach takes too long to run on the full test set). Over a variety of challenging transformations on MNIST and CIFAR10, our approach consistently achieves state-of-the-art performance. On all MNIST experiments, our certified robustness is substantially higher than DeepG, while attaining comparable clean accuracy. Furthermore, our verifier

Table 1: Comparison of network certification time and accuracy with the prior state-of-the-art on MNIST and CIFAR10. We denote $R(\varphi)$ a rotation of $\pm\varphi$ degrees; $T_u(\Delta u)$ and $T_v(\Delta v)$ a translation of $\pm\Delta u$ pixels horizontally and $\pm\Delta v$ pixels vertically, respectively; $Sc(\lambda)$ a scaling of $\pm\lambda\%$; $Sh(\gamma)$ a shearing of $\pm\gamma\%$; $C(\alpha)$ a contrast change of $\pm\alpha\%$; and $B(\beta)$ a brightness change of $\pm\beta$. Asterisk denotes that certification was measured on a subset of 100 test images.

| Dataset | Transformations | Training + Certification Method | Accuracy (%) | Certified (%) | Cert. Time per Image (s) | FGV Speedup |
|---------|-----------------|--------------------------------|--------------|---------------|--------------------------|-------------|
| MNIST | R(30) | PGD/A + DeepG | **99.1** | 86.0* | 19.12 | – |
| | | CGT + FGV | **99.1** | **94.2** | 0.00045 | 42623× |
| | $T_u(2)$, $T_v(2)$ | PGD/A + DeepG | 99.1 | 77.0* | 367.82 | – |
| | | CGT + FGV | **99.2** | **89.8** | 0.0090 | 40949× |
| | Sc(5), R(5), C(5), B(0.01) | PGD/A + DeepG | **99.3** | 34.0* | 155.24 | – |
| | | CGT + FGV | 99.1 | **92.6** | 0.0048 | 32563× |
| | Sh(2), R(2), Sc(2), C(2), B(0.001) | PGD/A + DeepG | **99.2** | 72.0* | 71.72 | – |
| | | CGT + FGV | 99.1 | **96.3** | 0.024 | 2933× |
| CIFAR10 | R(10) | PGD/A + DeepG | 71.2 | **65.0*** | 78.18 | – |
| | | CGT + FGV | **80.5** | 63.2 | 0.465 | 168× |
| | R(2), Sh(2) | PGD/A + DeepG | 68.5 | 39.0* | 18.92 | – |
| | | CGT + FGV | **70.1** | **51.0** | 0.263 | 72× |
| | Sc(1), R(1), C(1), B(0.001) | PGD/A + DeepG | **73.2** | 43.0* | 163.26 | – |
| | | CGT + FGV | 71.3 | **42.3** | 2.725 | 60× |

is several orders of magnitude faster. On CIFAR10, we achieve significantly better tradeoff between certified robustness and accuracy than the baseline for the first two cases, while obtaining similar results on the third transformation set; our certification time is considerably lower than DeepG. For the first and third CIFAR10 experiments, we attain 65% and 49% on the subset of 100 images used by DeepG (which are equal or better than their results), while the certifiability evaluated over the whole test set (presented in the table) is slightly lower. These results show that rather than using more precise abstractions on the input set to certify networks not trained to be provably robust, it is more effective to explicitly train networks to be certifiably robust and verify them with a less precise but faster verifier (with a large number of parameter splits). Using our custom loss formulation in Eq. 8 yields the best results on our verifier; Table 6 in Appendix E.1 shows that other existing $\ell_\infty$-based training methods (which cannot incorporate geometric bounds) do not perform as well.

## 5.2 SCALABILITY TO LARGER DATASETS AND MODELS

Our work can scale to larger datasets like Tiny ImageNet and a real-world autonomous driving dataset where previous approaches do not.

Table 2 presents our results on Tiny-ImageNet networks trained for the shear, scale, and rotation transformations. For both networks, we observe that our approach preserves a relatively high clean accuracy while obtaining substantial robustness guarantees. For context, in Xu et al. (2020) for $\ell_\infty$-robustness with $\epsilon = 1/255$, they attain 21.6% accuracy and 12.7% certified robustness on CNN7 and 27.8% accu-

Table 2: Certified geometric robustness and accuracy on Tiny ImageNet networks. Certification is performed with FGV.

| Network | Transforms | Acc. (%) | Cert. (%) | Cert. Time per Image (s) |
|---------|-----------|----------|-----------|--------------------------|
| CNN7 | Shear (2%) | 27.3 | 18.7 | 0.059 |
| | Scale (2%) | 26.1 | 15.2 | 0.057 |
| | Rotate (5°) | 26.0 | 13.1 | 0.285 |
| Wide ResNet | Shear (2%) | 35.5 | 25.7 | 0.214 |
| | Scale (2%) | 33.1 | 21.3 | 0.205 |
| | Rotate (5°) | 32.2 | 17.4 | 1.006 |

racy and 15.9% certified robustness on WideResNet. Certification time is still fast, even though the images are much higher-definition than MNIST and CIFAR10, showcasing FGV's scalability. We also see that our approach scales well to large state-of-the-art models like WideResNet.

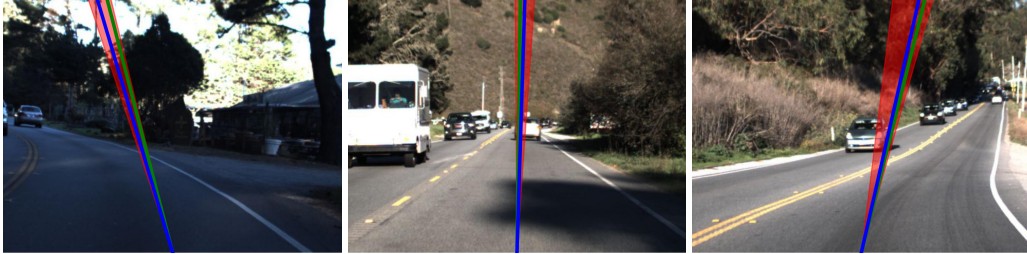

Figure 1: Visualization of standard and certified steering angles for the CGT-trained network on test-set driving scenes. Green line is the ground truth, blue line is the network prediction, and red hue is the certified bound on the network's prediction within $\pm 2°$ rotation of the input.

Next, we demonstrate, to the best of our knowledge, the first study of a provable defense in a real-world setting against realistic geometric transformations on the Udacity dataset. The neural network takes as input high-definition RGB $3 \times 66 \times 200$ images, and our task is to accurately predict the correct steering angle *and* tightly and provably bound the range of angles under geometric transformations. Specifically, we consider a transformation of $\pm 2°$ rotation.

While there is an inherent tradeoff between certified robustness and network performance on the previous datasets, we find that, surprisingly, here *we can scale our approach to obtain both high certifiability and improve network performance at the same time*. To demonstrate this phenomenon, we train a network with (1) just the regular mean-squared error loss and input data augmentation of $\pm 2°$ rotation (Regular), (2) the first method with the addition of dropout on linear layers (Dropout), and (3) our CGT formulation in Eq. 9.

Table 3: Mean absolute error and certified bounds (lower is better) for rotation range of $\pm 2°$ on self-driving networks.

| Training Method | Std. MAE | Cert. MAE | Cert. Width | Cert. Time per Image (s) |
|---|---|---|---|---|
| Regular | 6.07° | 97.56° | 180° | 0.11 |
| Dropout | **4.85°** | 96.65° | 178° | 0.12 |
| CGT | 5.36° | **8.05°** | **5.43°** | 0.11 |

Table 3 presents our results for the standard and certified errors of these training methods (lower is better). An interesting observation is that this task is prone to overfitting: we can observe that adding dropout significantly improves the standard error of the network. In this case, training with CGT's interval bounds actually acts as an effective regularizer that serves the dual purpose of combatting overfitting *and* enforcing certifiability. Training without CGT leads to trivial certification bounds (i.e., the certified steering angle can be anywhere in $\pm 90°$); conversely, training with CGT yields very tight certified bounds. Fig. 1 shows visualizations of the CGT-trained network's predictions and certified bounds.

### 5.3 ABLATION STUDIES

**Algorithm for Interval Interpolated Transformation.** In Appendix E.3, we compare Algorithm 1 to Semantify-NN's algorithm for computing interval rotation bounds. We show that our algorithm is orders of magnitude faster, hence enabling our scalable training and verification approaches.

**Effect of Parameter Split Size.** We train a CIFAR10 and Tiny ImageNet network with various sizes of the hyperparameter $\nu$ and discuss how it affects certified and clean accuracy in Appendix E.2.

## 6 CONCLUSION

We proposed Certified Geometric Training (CGT), a provable defense that leverages a novel fast geometric verifier to improve the deterministic certified robustness of neural networks with respect to geometric transformations. Our experiments across multiple datasets and perturbation sets showed that CGT consistently attains state-of-the-art certified deterministic geometric robustness and clean accuracy, while being highly scalable.

## ACKNOWLEDGMENTS

This research was supported in part by NSF Grants No. CCF-1846354, CCF-1956374, CCF-200888, CNS-2148583, CCF-2238079, USDA Grant No. AG NIFA 2021-67021-33449, a Sloan UCEM Graduate Scholarship, and a Qualcomm Innovation Fellowship.

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

## A  EQUATIONS FOR INTERPOLATED TRANSFORMATIONS

For each interpolated transformation, we present the equation for its inverse transform $T_\theta^{-1}$, which is used to instantiate Eq. 2.

*Rotation.* Parameterized by an angle $\varphi \in [0, 2\pi]$:

$$T_\varphi^{-1}(u, v) = \begin{bmatrix} \cos\varphi & \sin\varphi \\ -\sin\varphi & \cos\varphi \end{bmatrix} \begin{bmatrix} u \\ v \end{bmatrix} = \begin{bmatrix} u\cos\varphi + v\sin\varphi \\ v\cos\varphi - u\sin\varphi \end{bmatrix} \tag{10}$$

*Translation.* Parameterized by an amount of horizontal shift $\Delta u \in \mathbb{R}$ and an amount of vertical shift $\Delta v \in \mathbb{R}$:

$$T_{\Delta u, \Delta v}^{-1}(u, v) = \begin{bmatrix} u - \Delta u \\ v - \Delta v \end{bmatrix} \tag{11}$$

*Scaling.* Parameterized by a scaling factor $\lambda \in \mathbb{R}, \lambda > -1$:

$$T_\lambda^{-1}(u, v) = \begin{bmatrix} \frac{1}{1+\lambda} & 0 \\ 0 & \frac{1}{1+\lambda} \end{bmatrix} \begin{bmatrix} u \\ v \end{bmatrix} = \begin{bmatrix} u/(1+\lambda) \\ v/(1+\lambda) \end{bmatrix} \tag{12}$$

*Shearing.* Parameterized by a horizontal shearing factor $\gamma \in \mathbb{R}$:

$$T_\gamma^{-1}(u, v) = \begin{bmatrix} 1 & -\gamma \\ 0 & 1 \end{bmatrix} \begin{bmatrix} u \\ v \end{bmatrix} = \begin{bmatrix} u - \gamma v \\ v \end{bmatrix} \tag{13}$$

## B  INTERVAL BOUND PROPAGATION ABSTRACT TRANSFORMERS

Interval bound propagation (IBP) is a special case of linear relaxation-based perturbation analysis (LiRPA), where each neuron's bounds are hyperrectangles. Below, we give a recap on how to compute the bounds for affine (i.e., convolutional and fully connected) layers and monotonic activation functions from Gowal et al. (2019). For a more detailed discussion, we refer the reader to Gowal et al. (2019) and Xu et al. (2020).

For a neuron (or pixel) $\tilde{z} = [\underline{z}, \overline{z}]$, its bounds after applying an affine layer with weights $W$ and bias $b$ are computed as:

$$\tilde{z}_{out} = [\mu - r, \mu + r] \tag{14}$$

where $\mu = W\left(\frac{\overline{z}+\underline{z}}{2}\right) + b$ and $r = |W|\left(\frac{\overline{z}-\underline{z}}{2}\right)$.

For a neuron $\tilde{z} = [\underline{z}, \overline{z}]$, its bounds after applying a monotonic activation function $h\colon \mathbb{R} \to \mathbb{R}$ (e.g., ReLU) are computed as:

$$\tilde{z}_{out} = [h(\underline{z}), h(\overline{z})] \tag{15}$$

We can obtain final bounds on a network's outputs by composing the operations above and propagating intervals from the input layer to the output layer.

## C  ADDITIONAL DETAILS ON FAST INTERVAL INTERPOLATED TRANSFORMATIONS

### C.1  RUNNING EXAMPLE

We present a running example of Algorithm 1. Here, we consider applying a scaling of $\tilde{\lambda} = \pm 2\%$ to $3 \times 3$ images. Per Eq. 12, the inverse transform function for this perturbation is:

$$T_{\tilde{\lambda}}^{-1}(u, v) = \left( \frac{u}{1 + [-0.02, 0.02]}, \frac{v}{1 + [-0.02, 0.02]} \right) = \left( \frac{u}{[0.98, 1.02]}, \frac{v}{[0.98, 1.02]} \right)$$

We color code the diagrams below, so that information belonging to the same pixel location has the same color.

*Lines 2-4:* First, we create a meshgrid of $(u, v)$ coordinates so that the inverse transform of all coordinates can be computed in parallel. We obtain:

$$U = \begin{array}{|c|c|c|} \hline -1 & 0 & 1 \\ \hline -1 & 0 & 1 \\ \hline -1 & 0 & 1 \\ \hline \end{array} \text{ and } V = \begin{array}{|c|c|c|} \hline 1 & 1 & 1 \\ \hline 0 & 0 & 0 \\ \hline -1 & -1 & -1 \\ \hline \end{array}$$

*Line 5:* Now, we apply $T_{\tilde{\lambda}}^{-1}$ to each entry in $U, V$. Note that since $T_{\tilde{\lambda}}^{-1}$ produces interval coordinates, all subsequent operations need to be interpreted via interval arithmetic. We have:

$$\tilde{U}' = \frac{U}{[0.98, 1.02]} = \begin{array}{|c|c|c|} \hline [-1.02, -0.98] & [0, 0] & [0.98, 1.02] \\ \hline [-1.02, -0.98] & [0, 0] & [0.98, 1.02] \\ \hline [-1.02, -0.98] & [0, 0] & [0.98, 1.02] \\ \hline \end{array}$$

$$\tilde{V}' = \frac{V}{[0.98, 1.02]} = \begin{array}{|c|c|c|} \hline [0.98, 1.02] & [0.98, 1.02] & [0.98, 1.02] \\ \hline [0, 0] & [0, 0] & [0, 0] \\ \hline [-1.02, -0.98] & [-1.02, -0.98] & [-1.02, -0.98] \\ \hline \end{array}$$

*Lines 6-7:* We can now compute the bilinear interpolation grid (i.e., all interpolation distances). For each $(\tilde{u}', \tilde{v}')$ coordinate, we compute $\tilde{u}' - U$ and $\tilde{v}' - V$, which are both $3 \times 3$ interval matrices. As there are 9 inverse coordinates, we end up with two $9 \times 3 \times 3$ interval tensors $\tilde{U}_d$ and $\tilde{V}_d$ (where the ordering of the first dimension is according to row-major order of $\tilde{u}', \tilde{v}'$). The rest of the operations to compute the interpolation distances are all elementwise; we can thus obtain $\tilde{G} = \max(0, 1 - |\tilde{V}_d|) \odot \max(0, 1 - |\tilde{U}_d|)$, shown below. We omit writing entries that are zero (i.e., the degenerate interval $[0, 0]$).

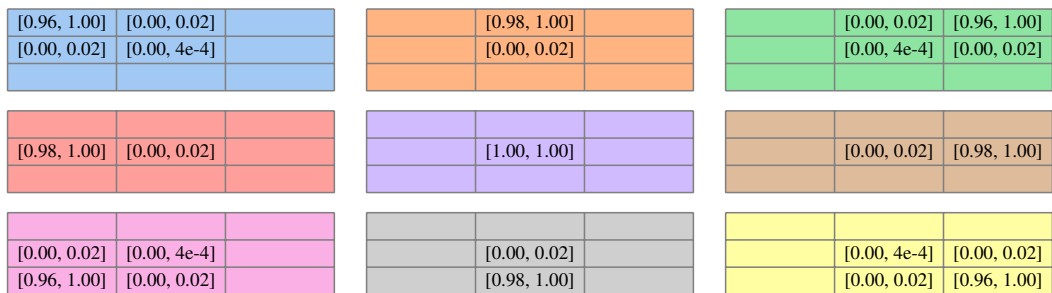

As discussed in Section 4.2, we can see that the region of interpolation is significantly different for each pixel, hence in order to parallelize the entire computation across all inverse coordinates, we must interpolate over the entire $3 \times 3$ range of $(u, v)$ coordinates for each $(\tilde{u}', \tilde{v}')$. However, doing so leads to a lot of sparsity in $\tilde{G}$, which we can now exploit. (With larger images, typically more than 99% of $\tilde{G}$ are zero entries; however, since the image dimension in this running example is small, the sparsity is not as great.)

*Lines 8-12:* We now convert $\tilde{G}$ to our custom sparse format before performing interpolation with image pixels, so that all zero-multiplications are eliminated. First, we count the number of nonzero entries in each inverse coordinate's matrix:

$$z = \begin{array}{|c|c|c|c|c|c|c|c|c|} \hline 4 & 2 & 4 & 2 & 1 & 2 & 4 & 2 & 4 \\ \hline \end{array}$$

Then, we flatten $\tilde{G}$ (in row-major order) and store it in a COO (coordinate) format. We thus obtain an interval vector $\tilde{w}$ that stores all the nonzero interpolation distances, along with integer vectors $r, c$ which, for each value of $\tilde{w}$, stores its corresponding row and column index:

$$\tilde{w} = \boxed{[0.96, 1.00]} \boxed{[0.00, 0.02]} \boxed{[0.00, 0.02]} \boxed{[0.00, 4e\text{-}4]} \boxed{[0.98, 1.00]} \boxed{[0.00, 0.02]} \boxed{[0.00, 0.02]} \boxed{[0.96, 1.00]} \boxed{[0.00, 4e\text{-}4]} \boxed{[0.00, 0.02]}$$

$$\cdots \boxed{[0.98, 1.00]} \boxed{[0.00, 0.02]} \boxed{[1.00, 1.00]} \boxed{[0.00, 0.02]} \boxed{[0.98, 1.00]} \boxed{[0.00, 0.02]} \boxed{[0.00, 4e\text{-}4]} \boxed{[0.96, 1.00]} \boxed{[0.00, 0.02]} \boxed{[0.00, 0.02]}$$

$$\cdots \boxed{[0.98, 1.00]} \boxed{[0.00, 4e\text{-}4]} \boxed{[0.00, 0.02]} \boxed{[0.00, 0.02]} \boxed{[0.96, 1.00]}$$

$$r = \boxed{0\;0\;1\;1\;0\;1\;0\;0\;1\;1\;1\;1\;1\;1\;1\;1\;1\;2\;2\;1\;2\;1\;1\;2\;2}$$

$$c = \boxed{0\;1\;0\;1\;1\;1\;1\;2\;1\;2\;0\;1\;1\;1\;2\;0\;1\;0\;1\;1\;1\;1\;2\;1\;2}$$

This concludes the procedure `MakeInterpGrid()`, and we are now ready to use this information to compute actual interpolated images. Consider a batch of 1000 $3 \times 3$ RGB images, $X \in [0,1]^{1000 \times 3 \times 3 \times 3}$. Now, consider the first channel of the first image, $X[0,0]$, which is a $3 \times 3$ matrix (whose values we randomly select for this example):

$$X[0,0] = \begin{array}{|c|c|c|} \hline .55 & .50 & .42 \\ \hline .53 & .49 & .51 \\ \hline .56 & .62 & .45 \\ \hline \end{array}$$

*Line 18:* We multiply each interpolation distance with the pixel value of $X[0,0]$ in the corresponding location; this will yield the values of all summands in Eq. 1 across all pixels. To accomplish this, we index the image according to the indices stored in $(r,c)$ and elementwise multiply these pixel values with $\tilde{w}$, obtaining $\tilde{S} = \tilde{w} \odot X[0,0,r,c]$:

$\tilde{S} =$ [0.53, 0.55] [0.00, 0.01] [0.00, 0.01] [0.00, 2e-4] [0.49, 0.50] [0.00, 0.01] [0.00, 0.01] [0.40, 0.42] [0.00, 2e-4] [0.00, 0.01]

$\cdots$ [0.52, 0.53] [0.00, 0.01] [0.49, 0.49] [0.00, 0.01] [0.50, 0.51] [0.00, 0.01] [0.00, 2e-4] [0.54, 0.56] [0.00, 0.01] [0.00, 0.01]

$\cdots$ [0.61, 0.62] [0.00, 2e-4] [0.00, 0.01] [0.00, 0.01] [0.43, 0.45]

*Line 19:* Finally, we sum the terms in $\tilde{S}$ that belong to the same pixel location. In the context of the diagram above, this is summing the entries in $\tilde{S}$ that are of the same color. In practice, this color information is encoded in the vector $z$, which stores, for each pixel location, the number of nonzero interpolation entries; hence, we break $\tilde{S}$ into contiguous chunks such that the length of chunk $i$ (where $0 \le i < 9$) is given by $z_i$, and sum all values in each chunk, obtaining:

$\tilde{X}'_f =$ [0.53, 0.57] [0.49, 0.51] [0.40, 0.44] [0.52, 0.54] [0.49, 0.49] [0.50, 0.52] [0.54, 0.58] [0.61, 0.63] [0.43, 0.47]

*Line 20:* This vector is then reshaped to a $3 \times 3$ matrix, yielding the final pixel values of $X[0,0]$ under transformation:

$$\tilde{X}' = \begin{array}{|c|c|c|} \hline [0.53, 0.57] & [0.49, 0.51] & [0.40, 0.44] \\ \hline [0.52, 0.54] & [0.49, 0.49] & [0.50, 0.52] \\ \hline [0.54, 0.58] & [0.61, 0.63] & [0.43, 0.47] \\ \hline \end{array}$$

Finally, we remark that this interpolation process is completely independent for each batch and channel. Therefore, to parallelize across multiple images and channels, we simply need to index across all batch and channel dimensions when computing $\tilde{S}$ (i.e., let $\tilde{S} = \tilde{w} \odot X[:,:,r,c]$).

## C.2 PADDING STRATEGY

To handle other padding techniques (e.g., replicating the border pixels of the in-frame part of the image), our core algorithm remains *completely unchanged* – all that is needed is a preprocessing step that pads the input image according to the desired padding strategy, and a postprocessing step that crops the output back to the original dimensions. We describe these steps in detail below.

Let us assume we have a batch of $H \times W$ images $X$, an interpolated transformation $T_{\tilde{\theta}}$, and a desired padding mode `strategy` by which to fill in the areas of the interpolated images with no source pixel. First, we determine the number of pixels by which to pad the original $H \times W$ images. To do so, we compute the meshgrid $(U, V)$ and the inverse coordinates $(\tilde{U}', \tilde{V}')$ as in ① in Algorithm 1. Then, the amount of padding required is $p = \lceil \max\{|\min \tilde{U}' - \min U|, |\max \tilde{U}' - \max U|, |\min \tilde{V}' - \min V|, |\max \tilde{V}' - \max V|\} \rceil$. In other words, the padding amount is the integer just greater than the maximum distance from a $u$ or $v$ coordinate on the border to its inverse counterpart; this ensures that after padding, none of the central $H \times W$ pixels will have a value of zero after interpolation. Now, we pad $X$ on all sides by $p$ pixels according to `strategy`. This padded batch of images can then be directly fed into Algorithm 1 to obtain a batch of transformed interval images $\tilde{X}'$. Finally, to obtain transformed images in the original dimensions, we take a central $H \times W$ crop of $\tilde{X}'$.

# D  ADDITIONAL EXPERIMENTAL DETAILS

## D.1  NETWORK ARCHITECTURES

We detail the network architecture(s) used for each dataset below. We express a convolutional layer as a 4-tuple of (number of filters, kernel size, stride, padding).

- **MNIST:** 2 conv layers $\{(32, 4, 2, 1), (64, 4, 2, 1)\}$ followed by 2 linear layers with $\{200, 10\}$ neurons. All layers are followed by a ReLU activation, except for the final output layer.
- **CIFAR10:** 3 conv layers $\{(32, 3, 1, 1), (32, 4, 2, 1), (64, 4, 2, 1)\}$ followed by 2 linear layers with $\{150, 10\}$ neurons. All layers are followed by a ReLU activation, except for the final output layer.
- **Tiny ImageNet:**
  - **CNN7:** 5 conv layers $\{(64, 3, 1, 1), (64, 3, 1, 1), (128, 3, 2, 1), (128, 3, 1, 1), (128, 3, 2, 1)\}$ followed by 2 linear layers with $\{512, 200\}$ neurons. Each conv layer is followed by a batch norm layer then a ReLU activation. The first linear layer is followed by a ReLU activation.
  - **WideResNet:** Let $Q_x(p_i, p_o, s)$ denote the output after feeding $x$ through this sequence of layers: batch norm with $p_i$ features, ReLU, conv$(p_o, 3, 1, 1)$, batch norm with $p_o$ features, ReLU, conv$(p_o, 3, s, 1)$; let $S_x(p_i, p_o, s)$ denote the output after feeding $x$ into a conv$(p_o, 1, s, 0)$. Then, we define a wide basic block as the function $W_x(p_i, p_o, s) = Q_x(p_i, p_o, s) + S_x(p_i, p_o, s)$. The architecture of the network is then: a conv layer $(16, 3, 1, 1)$, $W_x(16, 160, 1)$, $W_x(160, 320, 2)$, $W_x(320, 640, 2)$, batch norm, ReLU, average pooling with a $7 \times 7$ kernel, then finally a linear layer with 200 neurons.
- **Driving:** 5 conv layers $\{(24, 5, 2, 0), (36, 5, 2, 0), (48, 5, 2, 0), (64, 3, 1, 0), (64, 3, 1, 0)\}$ followed by 4 linear layers with $\{100, 50, 10, 1\}$ neurons. All layers are followed by a ReLU activation, except for the final output layer. For the Dropout network, we add a dropout layer with $p = 0.5$ after the first 3 linear layers.

## D.2  ADDITIONAL TRAINING DETAILS

**Training Schedule.**  We train the MNIST networks for 100 epochs with batch size 256, CIFAR10 networks for 120 epochs with batch size 128, Tiny ImageNet networks for 160 epochs with batch size 128 (CNN7) or 400 (WideResNet), and the self-driving network for 50 epochs with batch size 128. For the classifiers, we first train with *only* the cross-entropy loss during a warm-up period; we warm up for 15 epochs on MNIST and 30 epochs on CIFAR10 and Tiny ImageNet. For the self-driving network, we directly use Eq. 9 from the start. In order to ensure convergence for the loss, we linearly decay $\kappa$ from 1 to $\kappa_f = 0.5$ and employ a linear ramp-up schedule to slowly increase the value of $\nu$ from 0 up to a final parameter size of $\nu_f$; we ramp up across 50, 60, 80, and 50 epochs for MNIST, CIFAR10, Tiny ImageNet, and self-driving, respectively. We explain how to tune the hyperparameter $\nu_f$ and provide the values of $\nu_f$ for each experiment in Section D.4.

**Data Preprocessing and Augmentation.**  For all networks, we augment the input images during training according to the set of transformations to which we want to be robust. In addition, for CIFAR10 and Tiny ImageNet, we also perform random horizontal flips and select random crops of $32 \times 32$ with padding 4 (for CIFAR10) and random crops of $56 \times 56$ (for Tiny ImageNet). At test time, we use a central crop of $56 \times 56$ for Tiny ImageNet. For the self-driving dataset, we first crop the top of the $480 \times 640$ input images to $280 \times 640$, then resize them with bilinear interpolation to $66 \times 200$; we also perform random horizontal flips. For all datasets, we normalize input images according to the channel statistics from the train set immediately before the first network layer.

**Optimizer.**  We use the Adam optimizer (Kingma & Ba, 2015) across all networks. For MNIST and CIFAR10, we use an initial learning rate of $10^{-3}$, which we decay by 0.1 at the 80th and 100th epoch, respectively. For Tiny ImageNet, we use an initial learning rate of $5 \times 10^{-4}$, which we decay by 0.1 after 120 and 150 epochs. For self-driving, we use an initial learning rate of $10^{-3}$, which we decay by 0.1 after 20 and 40 epochs. For all networks, we clip gradients at an $\ell_2$-norm of 8.

Table 4: Interval sizes of geometric transformation parameters used during training and certification. For experiments with compositions of transformations, the sizes are specified in the same order as the transformations.

| Dataset | Perturbations | Perturbation parameter interval size | |
|---|---|---|---|
| | | Training ($2\nu_f$) | Certification ($\overline{\theta_k} - \underline{\theta_k}$) |
| MNIST | R(30) | 0.5 | 0.25 |
| | $T_u(2)$, $T_v(2)$ | $(0.1, 0.1)$ | $(0.05, 0.05)$ |
| | Sc(5), R(5), C(5), B(0.01) | $(1, 0.25, 5, 0.02)$ | $(0.5, 0.125, 5, 0.02)$ |
| | Sh(2), R(2), Sc(2), C(2), B(0.001) | $(0.5, 0.125, 0.5, 4, 0.002)$ | $(0.25, 0.0625, 0.25, 4, 0.002)$ |
| CIFAR10 | R(10) | 0.001 | 0.0002 |
| | R(2), Sh(2) | $(0.02, 0.05)$ | $(0.01, 0.025)$ |
| | Sc(1), R(1), C(1), B(0.001) | $(0.005, 0.005, 0.5, 0.002)$ | $(0.005, 0.005, 0.5, 0.002)$ |
| Tiny ImageNet | Sh(2) | 0.01 | 0.002 |
| | Sc(2) | 0.01 | 0.002 |
| | R(5) | 0.005 | 0.001 |
| Driving | R(2) | 0.004 | 0.001 |

Table 5: Average and maximum pixel interval widths of geometrically transformed images used for training (for CGT networks) and certification (for all networks).

| Dataset | Perturbations | Training Interval Width | | Certification Interval Width | |
|---|---|---|---|---|---|
| | | Average | Maximum | Average | Maximum |
| MNIST | R(30) | 0.019 | 0.234 | 0.010 | 0.124 |
| | $T_u(2)$, $T_v(2)$ | 0.041 | 0.334 | 0.022 | 0.181 |
| | Sc(5), R(5), C(5), B(0.01) | 0.043 | 0.356 | 0.031 | 0.223 |
| | Sh(2), R(2), Sc(2), C(2), B(0.001) | 0.022 | 0.253 | 0.015 | 0.153 |
| CIFAR10 | R(10) | $2.67 \times 10^{-4}$ | $8.53 \times 10^{-4}$ | $5.38 \times 10^{-5}$ | $1.71 \times 10^{-4}$ |
| | R(2), Sh(2) | $8.94 \times 10^{-3}$ | $2.82 \times 10^{-2}$ | $4.52 \times 10^{-3}$ | $1.42 \times 10^{-2}$ |
| | Sc(1), R(1), C(1), B(0.001) | $6.37 \times 10^{-3}$ | $1.26 \times 10^{-2}$ | $6.37 \times 10^{-3}$ | $1.26 \times 10^{-2}$ |
| Tiny ImageNet | Sh(2) | $1.21 \times 10^{-3}$ | $4.90 \times 10^{-3}$ | $2.44 \times 10^{-4}$ | $9.82 \times 10^{-4}$ |
| | Sc(2) | $2.39 \times 10^{-3}$ | $8.76 \times 10^{-3}$ | $4.81 \times 10^{-4}$ | $1.76 \times 10^{-3}$ |
| | R(5) | $2.15 \times 10^{-3}$ | $7.88 \times 10^{-3}$ | $8.63 \times 10^{-4}$ | $3.17 \times 10^{-3}$ |
| Driving | R(2) | $2.82 \times 10^{-3}$ | $1.51 \times 10^{-2}$ | $7.71 \times 10^{-4}$ | $4.00 \times 10^{-3}$ |

### D.3 Additional Certification Details

**Batch Size.** We use a batch size of 10,000, 10,000, 2,000, 400, and 3,000 during the certification of MNIST, CIFAR10, Tiny ImageNet CNN7, Tiny ImageNet WideResNet, and self-driving networks, respectively.

**Parameter Splits.** To ensure precise certification, we select parameter splits that are 1-5× smaller in width than those used during training. The certification configurations for each experiment can be found in Section D.4.

### D.4 Parameter Interval Sizes

In Table 4, we show the interval sizes of the geometric perturbation parameters that we use during training and certification for all experiments. In the Training column, we present the final (i.e., after ramp-up) interval size of each perturbation parameter used during training. Since we enforce a local geometric ball of up to $\pm\nu_f$ in Eqs. 8 and 9, the interval size of this ball is $2\nu_f$ (where $\nu_f$ is the final parameter interval size after ramp-up, as discussed in Section D.2). In the Certification column, we present the interval size of *each split* used during robustness certification. For experiments with

multiple perturbations, the ordering of the sizes corresponds with the ordering of the transformations in the Perturbations column.

We describe below the procedure to obtain the values of $\nu_f$ shown in Table 4:

1. For a given set of $n$ perturbations $P$ with interval parameter ranges $\tilde{\theta} = [\underline{\theta}, \overline{\theta}]$, we first start with an arbitrary $\nu_f$ such that $\nu_{f_i} < \overline{\theta}_i - \underline{\theta}_i$ for all $1 \leq i \leq n$.

2. Then, we uniformly sample 10 random scalar parameter values $\{\theta_k\}_{k=1}^{10}$ (where each $\theta_k \sim \mathcal{U}(\underline{\theta}, \overline{\theta})$ as in Eqs. 8 and 9) and compute, over all train set images $X \in [0, 1]^{N \times C \times H \times W}$, the maximum *pixel interval width* of each image under perturbation of $P$ with parameters $\theta_k \pm \nu_f$:

$$M_k = \{\max X_i'\}_{i=1}^N \text{ where } X' = P(X, \theta_k \pm \nu_f) \tag{16}$$

given $X_i'$ denotes the $i^{\text{th}}$ perturbed image in the train set. Essentially, in this step we are converting interval sizes in parameter space to pixel space to gauge the amount of over-approximation in the computed bounds (as it is the pixel's intervals that are ultimately propagated through the network). Now, we calculate the average maximum pixel interval width over all images and parameter samples, obtaining $\mu = \text{mean}\{\text{mean } M_k\}_{k=1}^{10}$.

We find that selecting $\nu_f$ such that $\mu$ is close to typical values of $\epsilon$ used in the $\ell_\infty$-norm setting (e.g., 0.2 and 0.4 for MNIST, $4/255$ and $8/255$ for CIFAR10) yields both accurate and certifiable networks. If $\mu > \epsilon$, then we reduce $\nu_f$ (hence reducing the amount of over-approximation) and recompute this step until $\mu \approx \epsilon$. Note that this tuning requires *no training*, and $\mu$ can be computed in *under a minute* for all the datasets that we consider.

3. Once appropriate values of $\nu_f$ have been determined from step (2), we can proceed to use them in training. We perform a 80-20 train-validation split of the train set, and use CGT to train a network to completion. If the validation accuracy and robustness are sufficiently high, then no further tuning is required. Else, we reduce $\nu_f$ and repeat this step. We only had to do this step a few times, as the calculation of the pixel widths already provided a good heuristic for network performance.

After determining the appropriate parameter sizes during training time, selecting the certification split sizes is straightforward. We empirically find that selecting parameter sizes that are $2\times$ smaller than those used during training yields a good balance between certification rate and runtime. If faster certification time is desired, one can use the same parameter sizes at both training and certification time (as we do in the CIFAR10 experiment with scaling, rotation, contrast, and brightness). If higher certification rate is desired, one can use parameter sizes that are even smaller, at the cost of higher runtime (e.g., we find $5\times$ to be effective for Tiny ImageNet, while using parameter sizes that are even smaller has diminishing returns and does not appreciably increase certification rate).

Table 5 shows the maximum pixel interval width during training and certification for each experiment. Additionally, we also show the average pixel interval width (i.e., Eq. 16 with the max operation replaced by mean) to demonstrate that geometric transformations produce highly nonuniform bounds, further motivating why supplying precise geometric bounds during training is crucial to achieving networks that are both accurate and certifiable.

## E    ADDITIONAL RESULTS

### E.1    TRAINING METHODS

To demonstrate the benefit of our loss function, we compare CGT with three baseline training methods: (1) data augmentation, denoted Augment.; (2) augmentation with PGD (Madry et al., 2018), denoted PGD+A; and (3) augmentation with IBP (Gowal et al., 2019), denoted IBP+A. We select these baselines because, to the best of our knowledge, they are the *only* training methods utilized in existing works on deterministic geometric robustness (as no work incorporates precise geometric regions into training as we do). For (1), the loss function is simply the cross-entropy loss, with input images augmented according to the perturbations we are trying to certify. For (2), we use PGD to generate adversarial examples of the augmented images. For (3), we employ the IBP loss, with interval $\epsilon$-balls placed around the augmented images.

Table 6: Comparison of training methods on the MNIST translation (left) and CIFAR10 rotation (right) benchmarks.

| Training Method | Accuracy (%) | Certified (%) | Time per Epoch (s) | Max GPU Mem. (MB) | Training Method | Accuracy (%) | Certified (%) | Time per Epoch (s) | Max GPU Mem. (MB) |
|---|---|---|---|---|---|---|---|---|---|
| CGT | 99.2 | 89.8 | 3.25 | 116.3 | CGT | 80.5 | 63.2 | 6.06 | 236.2 |
| IBP+A | 98.5 | 82.6 | 2.53 | 116.2 | IBP+A | 62.2 | 46.4 | 5.44 | 235.3 |
| PGD+A | 99.4 | 0 | 16.6 | 67.07 | PGD+A | 76.7 | 38.8 | 7.48 | 83.26 |
| Augment. | 99.4 | 0 | 1.50 | 64.76 | Augment. | 83.7 | 0.04 | 5.13 | 74.38 |

Table 6 shows these comparisons on the MNIST translation and CIFAR10 rotation benchmarks. For the PGD and IBP-based baselines, we select $\epsilon = 0.1$ and $\epsilon = 2/255$ for MNIST and CIFAR10, respectively, which are commonly used in the literature (Balunovic et al., 2019; Gowal et al., 2019; Zhang et al., 2020). For PGD, we use a step size of 0.005 and 40 steps for MNIST and a step size of 0.002 and 7 steps for CIFAR10. When calculating the time and memory statistics for CGT and IBP+A, we only consider the epochs after warm-up (i.e., when we start computing the interval robustness loss term). We can observe that our approach, which can supply precise bounds that correspond to the geometric perturbations we are trying to certify, is able to attain the best certified accuracy while maintaining high clean accuracy and only using slightly more time and memory compared to the other approaches.

## E.2 EFFECT OF HYPERPARAMETER $\nu$

Table 7: Variation of clean and certified accuracy as a function of $\nu_f$. $c$ is the certification interval size, $\nu_o$ is the value of $\nu_f$ used in our main results, and $g = \nu_o - c$.

| Benchmark | Accuracy Type | $\nu_f$ | | | | |
|---|---|---|---|---|---|---|
| | | $c$ | $c + \frac{g}{2}$ | $\nu_o$ | $\nu_o + \frac{g}{2}$ | $\nu_o + g$ |
| CIFAR10 | Certified | 45.6 | 49.8 | 51.0 | 51.7 | 50.8 |
| R(2), Sh(2) | Clean | 73.5 | 71.8 | 70.1 | 69.8 | 66.6 |
| Tiny ImageNet | Certified | 9.4 | 13.6 | 15.2 | 16.0 | 16.5 |
| Sc(2) | Clean | 28.5 | 26.9 | 26.1 | 25.4 | 25.2 |

As the size of the local geometric ball (controlled by $\nu$) is a key training hyperparameter of our loss function in Eq. 8, we conduct an ablation study on a CIFAR10 network and Tiny ImageNet CNN7 on how varying $\nu_f$ (i.e., the final parameter interval size after ramp-up, as discussed in Section D.2) affects the certified and clean accuracies of CGT-trained networks. Table 7 presents these results. In addition to the original value of $\nu_f$ used in our benchmarks in Sections 5.1 and 5.2 (denoted $\nu_o$), we select four additional settings for $\nu_f$ and train with CGT for each of these values. We choose the smallest $\nu_f$ to be equal to the interval size at certification time, $c$, and the largest $\nu_f$ to be equal to $c$ plus two times the difference between $\nu_o$ and $c$. We certify all networks with the original size of $c$.

We can observe that generally, larger values of $\nu_f$ lead to increased certified accuracy, at the cost of decreased clean accuracy. CGT thus allows one to explicitly tune the tradeoff between these metrics by varying $\nu_f$. The notable exception to this trend is the CIFAR10 network with the largest $\nu_f$; in this case, using a $\nu_f$ that is excessively large actually decreases certified robustness (since the clean accuracy drops considerably). There are diminishing returns to using larger intervals during training: while the certified accuracy increases significantly when using $\nu_f = \nu_o$ compared to $\nu_f = c$, further increasing the size to $\nu_f = \nu_o + g/2$ or above only slightly increases certifiability.

## E.3 SPEED OF INTERVAL INTERPOLATED TRANSFORMATION ALGORITHM

We demonstrate why FGV is a necessary component of our framework. Table 8 presents the comparison of Algorithm 1 to the state-of-the-art algorithm for computing interval geometric bounds in Semantify-NN (Mohapatra et al., 2020), which is sequential and CPU-only. To compute interval rotation bounds for a batch of training images (256 for MNIST and 128 for CIFAR10 and Tiny ImageNet), we are 8,000-20,000× faster. Our end-to-end speedup when combining both the input

Table 8: Time to compute interval rotation bounds and network outputs for one batch of images.

| Dataset | Time to Compute Bounds (s) | | Our Speedup | Time to Propagate Bounds (s) | End-to-End Speedup |
|---------|--------------|--------|-------------|-------------|---------|
| | Semantify-NN | Ours | | | |
| MNIST | 33.10 | **0.0029** | 11250× | 0.0037 | 4974× |
| CIFAR10 | 22.81 | **0.0026** | 8645× | 0.0043 | 3293× |
| Tiny ImageNet | 62.83 | **0.0030** | 20610× | 0.0179 | 3003× |

Table 9: Time to compute interval rotation bounds for one batch of images, with our method running on the CPU and with batch size 1.

| Dataset | Semantify-NN (s) | Ours CPU (s) | Our Speedup |
|---------|--------|--------|--------|
| MNIST | 33.10 | **0.25** | 135× |
| CIFAR10 | 22.81 | **0.11** | 200× |
| Tiny ImageNet | 62.83 | **0.28** | 221× |

geometric perturbation bound computation and the propagation of these bounds through the network is 3,000-5,000×. For Tiny ImageNet, we use the CNN7 network in this experiment. These results show that without Algorithm 1, the overhead during training is too high to use our proposed loss functions in Eqs. 8 and 9.

As an ablation study, we run our algorithm under the same conditions as Semantify-NN's – on the CPU and with a batch size of 1. As seen in Table 9, our runtime is still 100-200× faster than Semantify-NN's. These results show that our speedup is not only due to the ability to leverage GPU hardware, but also due to fundamental algorithmic improvements that allow the computation of interpolated transformations to be efficiently parallelized.

## E.4 Training Statistics

Table 10 shows the training statistics of runtime per epoch and GPU memory usage for all of our networks. When calculating the time and memory statistics, we only consider the epochs after warm-up (i.e., when we start computing the interval robustness loss term).

Table 10: Training runtime and GPU memory usage for all our benchmarks.

| Network | Perturbations | Time per Epoch (s) | Max GPU Memory (MB) |
|---------|---------------|--------------------|---------------------|
| MNIST | R(30) | 3.46 | 116.3 |
| | $T_u(2)$, $T_v(2)$ | 3.25 | 116.3 |
| | Sc(5), R(5), C(5), B(0.01) | 3.46 | 116.3 |
| | Sh(2), R(2), Sc(2), C(2), B(0.001) | 3.53 | 116.3 |
| CIFAR10 | R(10) | 6.06 | 236.2 |
| | R(2), Sh(2) | 6.02 | 236.3 |
| | Sc(1), R(1), C(1), B(0.001) | 6.11 | 237.3 |
| Tiny ImageNet CNN7 | Sh(2) | 48.8 | 4362 |
| | Sc(2) | 48.9 | 4362 |
| | R(5) | 48.9 | 4361 |
| Tiny ImageNet WideResNet | Sh(2) | 98.2 | 34143 |
| | Sc(2) | 98.1 | 34143 |
| | R(5) | 98.8 | 34143 |
| Driving | R(2) | 11.7 | 3428 |

