# OpenReview forum: "Provable Defense Against Geometric Transformations"
_ICLR.cc/2023/Conference — ICLR 2023 notable top 25%_

### Official Review · Reviewer_XMPM · 2022-10-15

**Confidence:** 2
**Correctness:** 4
**Technical Novelty And Significance:** 3
**Empirical Novelty And Significance:** 3
**Recommendation:** 6

**Clarity, Quality, Novelty And Reproducibility:**

The paper is clear, of good quality. The novelty is fair. Code is not provided, thus it is a little questionable that the paper can be reproduced.

**Strength And Weaknesses:**

Strength:
1. This paper is the first one that uses GPU accelerated training for certified robustness against geometric transformations.

2. The paper conducts an extensive study for certifying different models and datasets.



**Summary Of The Paper:**

This paper certifies the robustness of the neural network under geometric transformations. While L2 norm bounded certification has been used in training to further improve the certification bound, no one has used this in geometric transformations. Following the established GPU accelerated certification method, this paper can speed up the geometric robustness.

**Summary Of The Review:**

I do not work in the certification of neural networks, the results read solid in speed up and improving the geometric transformation certification. Thus I recommend for acceptance.

---

> ### Author Response · Authors · 2022-11-14
> **Response to Reviewer XMPM**
>
> Thank you for taking the time to review our paper and for your positive comments! We have made our code and trained networks available in the supplementary material. Please let us know if there is anything else we can address.

---

> > ### Comment · Reviewer_XMPM · 2022-12-05
> > **Thanks for your response.**
> >
> > The code is provided for reproducibility in the rebuttal. I keep my rate for acceptance.

---

### Official Review · Reviewer_56ik · 2022-10-23

**Confidence:** 4
**Correctness:** 4
**Technical Novelty And Significance:** 2
**Empirical Novelty And Significance:** 3
**Recommendation:** 6

**Clarity, Quality, Novelty And Reproducibility:**

The paper is well structured and easy to follow.
The novelty, to the best of my knowledge, lies in the parallelization of geometric perturbations that rely on interpolation operations. The proposed training loss is however straightforward and the certification is performed using auto LiRPA.
The authors promise that the codes will be made available upon publication.


**Strength And Weaknesses:**

It is an interesting idea to improve the scalability of the formal verification algorithms by accelerating / parallelizing perturbations. The appendices have also provided some additional insights.
Weaknesses:

1. The algorithm 1 as a core contribution would perhaps need more motivation and explanation. For instance, one could further modularize the big chunks of the algorithm into smaller functions with better defined purposes and input/output. Furthermore, it is not clear to me how this algorithm is better parallelizable than the standard methods. The paper could have been much stronger if this point had been highlighted. But since I'm no expert in algorithm efficiency and have probably missed certain aspects, I'm open to other reviewers' opinion.

2. To be more self-contained, perhaps one could include a small recap of the LiRPA.

3. I’m wondering why PGD+DeepG is expected to produce SoTA performances, since PGD is gradient based and DeepG geometric. The authors might want to include references that also make use of / explain this specific design.


**Summary Of The Paper:**

This paper introduces a framework to improve and certify robustness against certain geometric perturbations. The main contribution lies in the parallelization of geometric transformations and their inverses. These improvements enable efficient training as proposed in the paper, as well as efficient LiRPA verification. The conducted experiments demonstrate speedup up to multiple orders of magnitude.

**Summary Of The Review:**

 In general this is a sound paper with an interesting idea but limited novelty. This could be improved, however, if the main contribution, the parallelization, could be elaborated.

---

> ### Author Response · Authors · 2022-11-14
> **Response to Reviewer 56ik**
>
> Thank you for taking the time to review our paper and for providing us with valuable, constructive feedback and insightful questions – we really appreciate it! We have revised the paper according to your advice (with all changes in blue), and our responses to your questions are as follows.
>
> **The algorithm 1 as a core contribution would perhaps need more motivation and explanation. For instance, one could further modularize the big chunks of the algorithm into smaller functions with better defined purposes and input/output.**
>
> Thank you for these suggestions. We have updated the motivation and explanation (in Section 4.2, pages 5 and 6) to more clearly describe the reason why existing work has been unable to leverage parallelism. We also hope that the response below is helpful in providing further clarification. Additionally, we have added circled numbers and braces to clearly group chunks of our algorithm by functionality (here, we could not directly modularize the function due to space limitations).
>
> **Furthermore, it is not clear to me how this algorithm is better parallelizable than the standard methods. The paper could have been much stronger if this point had been highlighted. But since I'm no expert in algorithm efficiency and have probably missed certain aspects, I'm open to other reviewers' opinion.**
>
> For each pixel location $(i, j)$, the amount and location (i.e., structure) of neighboring pixels that contribute a non-zero term to that pixel’s interpolation grid can be drastically different (as we show graphically in Appendix C.1, center of page 13). Existing techniques like Algorithm 1 in Singh et al., 2019 and the method in Mohapatra et al., 2020 only interpolate over the non-zero terms specifically; this computation cannot be parallelized due to the aforementioned structural difference at each pixel. We restructure the computation such that the interpolated region is identical across all pixel locations, which is what allows us to parallelize the computation of the interpolation grid.
>
> However, naively using this interpolation grid to compute interpolated images leads to a requirement of having to then multiply with a large sparse tensor that is typically 0 at more than 99% of the entries. Multiplying by such a large tensor directly is prohibitively expensive (both in terms of memory and runtime), and thus an important part of our contribution is developing a custom sparse tensor data structure (specific to our problem) that allows us to realize the proposed parallel computation in practice.
>
> We have updated the writing under the “Interpolation Grid and Exploiting Sparsity” subheading of Section 4.2 to further clarify these aforementioned points.
>
> **To be more self-contained, perhaps one could include a small recap of the LiRPA.**
>
> Thank you for the helpful suggestion; we added an Appendix B (which we refer the reader to in Section 3.2) that provides a recap on how bounds are propagated through the neural network.
>
> **I’m wondering why PGD+DeepG is expected to produce SoTA performances, since PGD is gradient based and DeepG geometric. The authors might want to include references that also make use of / explain this specific design.**
>
> We would like to clarify that the networks attaining SoTA performances in Balunovic et al., 2019 are trained via PGD *in combination with data augmentation* (according to the set of geometric transformations that one would like to be robust against); we have changed the usage of “PGD” to “PGD/A” in Table 1 to make this point clear (where PGD/A denotes PGD with Augmentation). Among several training techniques (using just data augmentation, using interval bound propagation with data augmentation, etc.), Balunovic et al. established that PGD/A produced networks that have both the best accuracy *and* the best (deterministic) certifiable geometric robustness on their verifier.
>
> We have made our code available in the supplementary material.
> Thank you for your review once again, and please let us know if there is anything else we can address.

---

### Official Review · Reviewer_2P26 · 2022-10-24

**Confidence:** 5
**Correctness:** 4
**Technical Novelty And Significance:** 3
**Empirical Novelty And Significance:** 4
**Recommendation:** 8

**Clarity, Quality, Novelty And Reproducibility:**

## Quality
The paper achieves great empirical results, on an important problem, by proposing a simple solution.

## Clarity
The paper is very clear in its explanation of their algorithm, and presents itself correctly in the context of the existing literature.

## Originality
The idea is relatively simple, combining the concept of certified training with the specification of robustness to geometric transformation, but the solution proposed to enable this combination is definitely novel.

## Minor note:
- In Algorithm 1, I assume that line 17 should be a call to the `MakeInterpGrid` procedure defined above?


**Strength And Weaknesses:**

# Strength
The problem studied is quite an important one as geometric perturbations are a more realistic specifications than $\mathcal{L}_p$ bounds.
The paper is clearly written, with proper background being given for the problem studied (geometric perturbations, robust objectives), which makes it easy to follow along. The annotated Algorithm 1 is easy to follow, allowing for easy re-implementation by other people, and the running example of Appendix B is also useful for understanding.
The empirical results are quite convincing, and evaluated on a wide range of datasets and specification.


# Weakness - Things to improve.
- There is a lack of precision as to how bounds are propagated through the network. The authors describe very clearly how to propagate bounds through the geometric transformation, but then only describe that they use auto_lirpa to propagate through the network, without describing which algorithm they are using. The text after Definition 5 seems to indicate that this might be IBP, but it would be good to be clear on whether that's the case (is it the same both at training and verification time?)
- This is only a suggestion, but after reviewing the paper, I think what one of the things that the authors should highlight more is how different their method is from the most comparable works (Balunovic et al.) and (Mohapatra et al.). Both of these work go through the trouble of deriving bounds that are **explicitly dependent on $\theta$**, so that they can propagate linear bounds that are dependent on $\theta$. This paper completely circumvents that problem by using worse bounds which do *not* depend on $\theta$. It seems to me like this is an integral part of what makes things so efficient: given that you're going to chop up the input space in such tiny sub-intervals anyway, there is no reason to have super representative bounds.
- More generally, I think that the ablation study in 5.3 is the weakest part of the paper. The speedup number presented are impressive, but it seems to me that the comparison is a bit unfair, if the baseline method is not batched and operates on CPU rather than GPU. By all means, the author should provide these results but they also should include a runtime for their method under similar conditions (batch size of 1, running on CPU). I think that these would highlight clearly what the contribution is (parallelizing, vectorizable). As it is, this ablation study is not ablating anything, it's just comparing to a baseline. I'll clarify that even if the severely handicapped version of their algorithm is not outperforming Semantify-NN, that is still an acceptable result and that highlighting what choices need to be made to enable certified training is very interesting.
- I think the authors are missing a more detailed discussion of the verifiable robustness to rotation of (Singh et al.,  2019). It is cited as verifying the robustness against norm-based adversarial perturbations, but that paper actually also contains discussion of robustness to rotation, using exactly the same type of bounding through adversarial transformation.

**Summary Of The Paper:**

This paper deals with training neural network to be robust to geometric transformations of the inputs. Here, robustness is taken to mean *certified* robustness, which can be formally proven, even in the worst case scenario.

The contribution are two-fold:
- Improvements in terms of speed of evaluating bounds on the output of the network given that the input is subject to geometric transformations.
- Using the speed-up in verification, it becomes feasible to include the robust loss into the training objective, in order to train the network to be certifiably robust, which will make it easier to verify.

The speed up of the verification is achieved by:
- Relying on input splitting to only consider *small* perturbations, such that for each pixel, only a small number of original image pixels can contribute to the interpolation. This has two impacts: the bounds produced are tighter (because the bound is taken over only a small number of pixel), and the transformation can be represented using a very sparse representation, which can make it efficient.
- Batching. During training, the sampled transformation range is applied to *all* of the elements in the batch (it's not a different transformation per sample), which means that the computation of the interpolation coefficients is amortized over the batch. I assume that the same is done at verification time, where a batch of test samples are verified at the same time, and the same "slice" of the input domain is verified for all the images at the same time.



**Summary Of The Review:**

The paper provides a good solution to enable fast propagation of bounds through geometric transformations at training time, enabling certified training against geometric transformation. This simple idea, which is well explained and thoroughly evaluated leads to good empirical results.

---

> ### Author Response · Authors · 2022-11-14
> **Response to Reviewer 2P26**
>
> Thank you for taking the time to review our paper and for providing us with valuable, constructive feedback and insightful questions – we really appreciate it! We have revised the paper according to your advice (with all changes in blue), and our responses to your questions are as follows.
>
> **The authors describe very clearly how to propagate bounds through the geometric transformation, but then only describe that they use auto_lirpa to propagate through the network, without describing which algorithm they are using. The text after Definition 5 seems to indicate that this might be IBP, but it would be good to be clear on whether that's the case (is it the same both at training and verification time?)**
>
> The reviewer is correct that we use IBP to propagate our computed geometric bounds through the network – both during training and verification time. We have clarified this in the text below Definition 5 and at the end of Section 4.
>
> **Both of these works go through the trouble of deriving bounds that are explicitly dependent on $\\theta$, so that they can propagate linear bounds that are dependent on $\\theta$. This paper completely circumvents that problem by using worse bounds which do not depend on $\\theta$. It seems to me like this is an integral part of what makes things so efficient: given that you're going to chop up the input space in such tiny sub-intervals anyway, there is no reason to have super representative bounds.**
>
> We agree with the reviewer and have updated Section 3.3 (just before Definition 2) to emphasize this. Indeed, it takes a long time both to compute and propagate DeepG’s (Balunovic et al., 2019) polyhedral bounds due to every pixel’s constraints being explicitly dependent on the parameters $\\theta$. Mohapatra et al., 2020 also use worse interval bounds that do not depend on $\\theta$. However, their method of computing the geometric perturbation bounds cannot be parallelized and run on the GPU as ours can, which is why our verification is much more efficient. Consequently, our approach is also more precise: because our algorithm is much faster, we can afford to rerun it using a large number of splits, reducing over-approximation and thereby increasing verification precision.
>
> **More generally, I think that the ablation study in 5.3 is the weakest part of the paper. The speedup number presented are impressive, but it seems to me that the comparison is a bit unfair, if the baseline method is not batched and operates on CPU rather than GPU. By all means, the author should provide these results but they also should include a runtime for their method under similar conditions (batch size of 1, running on CPU).**
>
> We agree with the reviewer, and thank them for bringing this to our attention. We ran our algorithm as the reviewer suggested, with a batch size of 1 and on the same CPU as in the paper (which has 24 cores and 2 threads per core). In this setting, our approach is still 100-200$\\times$ faster, showing that our contribution’s speedup is not only due to the ability to leverage GPU hardware, but also due to fundamental algorithmic improvements. Due to space limitations, we have moved these results to Appendix E.3 Tables 8 and 9 (which we refer the reader to in Section 5.3).
>
> **I think the authors are missing a more detailed discussion of the verifiable robustness to rotation of (Singh et al., 2019).**
>
> The results of DeepPoly are strictly subsumed by the results of DeepG: both use the same verifier for propagating bounds through the network, but DeepG’s abstraction of the geometric perturbation input region is much more precise. In fact, the approach of Singh et al., 2019 is the baseline considered in DeepG. We have updated Section 2 (Related Work) to clarify this.
>
> **In Algorithm 1, I assume that line 17 should be a call to the MakeInterpGrid procedure defined above?**
>
> Since the interpolation grid for a given set of transformations and parameters only has to be computed *once*, we define the `Interpolate` function as taking in a precomputed grid $G$ (obtained from a previous call to `MakeInterpGrid`), instead of directly including a call to `MakeInterpGrid` on line 17. We have updated the text in the subsection “Obtaining Interpolated Images” in Section 4.2 to make this clear.
>
>
> Thank you for your review once again, and please let us know if there is anything else we can address.

---

### Official Review · Reviewer_Z6j5 · 2022-10-25

**Confidence:** 4
**Correctness:** 3
**Technical Novelty And Significance:** 4
**Empirical Novelty And Significance:** 4
**Recommendation:** 8

**Clarity, Quality, Novelty And Reproducibility:**

### Clarity
Well written overall, with good flow and clear definitions.

Some questions:

Q: How do we define the output for geometric transformations (any rotation, any translation, shrinking, any shearing) that move some parts of the image out of the frame and leave some areas of the frame with no corresponding source pixel in the image? This does not have a bearing on your method per se, but doesn't seem to be defined either in the main body or the Appendix.

Q: Are interpolated transformations bijective? Otherwise, how do you handle this? My toy example are the following matrices:

```
1 0
0 1

0 1
1 0
```

When 'rotated' by 45 degrees, they seem to yield the same output:

```
0.5 0.5
0.5 0.5
```

When rotating this output matrix by another 45 degrees, which of the original matrices should you obtain?

Again - this is not necessarily an issue with your method (but it does complicate checking that the math is correct)

### Quality

Improves on the state of the art by several orders of magnitude in runtime (for a problem that is of interest to the community) and matching or improving on certified accuracy.

### Novelty

Clearly novel verifier, and novel loss (albeit one that is a variation on existing robust losses).

### Reproducibility

The evaluation section (and additional details in the Appendix) provides all the information necessary to reproduce the results. (For example, the work specifies the exact hardware used and the architecture of the networks verified).

---

Separately from the comments above, a minor nit: I don't think that the (low probability) susceptibility of probabilistic guarantees to false negatives for verification is so serious to be 'not permissible'. What's a real-world use case under which such false negatives would cause a problem? Some thoughts:

- In the case of offline evaluation on a dataset, the low rate of false negatives should not be a problem.
- In the case of determining online for a particular scene (say, validating stop sign detection in the driving context), having a deterministic method that verifies at a low rate that the stop sign detected is robust to adversarial examples seems just as bad as a probabilistic method that verifies at a high rate ...

Perhaps I'm missing something here. I'd be interested to hear your thoughts


**Strength And Weaknesses:**

Well written paper with all the details required to reproduce the result. Improves verification runtime by several orders of magnitude while matching or improving on robust accuracy, via a novel GPU-accelerated verifier.

A minor concern I have is around the hyperparameter $\nu$. The schedule of values for these hyperparameter are provided in the Appendix, but it's not clear to me how the work arrives at these values in the first place. Referencing Table 5, $\nu$ does not have a clear relationship to the size of the original domain. If hyperparameter tuning was used to determine the appropriate value of $\nu$, this should be reflected in the main body - since that would increase the amount of time actually required for training and verification.

**Summary Of The Paper:**

The paper introduces a novel GPU-optimized verifier (FGV) for geometric transformations [1]. This verifier enables training using a sampling-based [2] robust classification loss leading to robust networks that are efficiently verifiable with the FGV. For small datasets (MNIST / CIFAR10), the proposed algorithm is orders of magnitude faster with comparable or better certified accuracy when compared to the state of the art. It also demonstrates non-trivial certification results on larger datasets (Tiny-ImageNet/Udacity self-driving car) for the first time.

[1] Rotations, translations, scaling, shearing, changes in contrast and changes in brightness
[2] Rather than computing the robust loss across the full input domain, the method computes the worst-case loss for a small subset of the input domain.

**Summary Of The Review:**

This paper is a clear accept. It introduces a novel GPU-accelerated verifier FGV for geometric transformations (the idea of using the GPU for verification is not novel, but applying it to geometric transformation is), allowing for a certifiable training approach that results in networks that are robust and verifiable via FGV. My only concern is that there does not seem to be a principled way to select the domain-splitting hyperparameter $\nu$; this may be something that has to be selected on a dataset-by-dataset basis.

---

> ### Author Response · Authors · 2022-11-14
> **Response to Reviewer Z6j5 (Part 2 of 2)**
>
> **Separately from the comments above, a minor nit: I don't think that the (low probability) susceptibility of probabilistic guarantees to false negatives for verification is so serious to be 'not permissible'.**
>
> We agree with the reviewer and have revised Section 2 (Related Work) to better reflect the limitations of probabilistic verification approaches. We also clarify that we intended the term “false negative” to refer to cases when the verifier falsely “verifies” an adversarial region to be robust, leading to such an adversarial example being undetected; we have removed the use of this term and directly stated this in the revised paper.
>
> - We agree with the reviewer that the low rate of false negatives is not a problem in the offline setting. In the online setting, we agree that while the uncertainty of probabilistic approaches is not so serious to be “not permissible,” there are still certain drawbacks: (1) using a smoothed network introduces randomness into the inference process (in addition to large runtime overhead, stated below) and (2) as mentioned above, some adversarial examples may be falsely certified as robust (and hence go undetected). In safety-critical settings, these concerns may be undesirable.
> - Probabilistic guarantees are obtained over a *smoothed* version of a base network, which at inference time requires sampling (i.e., repeatedly evaluating) the network up to 10,000 times per image, introducing orders of magnitude of runtime overhead (Fischer et al., 2020). In contrast, we certify the original network and thus this network incurs no overhead during inference.
>
> Thank you for your review once again, and please let us know if there is anything else we can address.

---

> > ### Comment · Reviewer_Z6j5 · 2022-11-15
> > **On probabilistic guarantees**
> >
> > Thanks for the update - I think the new paragraph is much more accurate in describing the situation :)
> >
> > Out of curiosity: you point out that techniques like randomized smoothing impose significant inference overhead at runtime, which means that a (fast) verification technique (like yours) could have a significant advantage. How long does ~10,000 inferences on the CIFAR-10 networks you've trained compare to the ~2.7s certified time per image for your most complex set of transformations? (I understand that this may be like comparing apples with oranges if we don't currently have a way to train networks robust to that particular set of transformations, but wanted to get a sense for orders of magnitudes).

---

> > > ### Author Response · Authors · 2022-11-17
> > > **Overhead of randomized smoothing**
> > >
> > > Thank you for the question: we would first like to clarify that the inference overhead of randomized smoothing is not just in terms of runtime, but also memory (we have revised the Related Work section accordingly).
> > >
> > > Many safety-critical applications require fast, on-the-fly inference on resource-constrained devices. Consider a standard scenario where we first obtain distributional certification guarantees on the CIFAR10 test set for a network that we would like to deploy in a real-time application: here, certification (using either a smoothing-based approach or our technique) is done *before* inference. At deployment time, running one inference on the trained networks for a single image takes 0.00065 seconds and 5.3 MB GPU memory. Running 10,000 inferences (as in randomized smoothing to maintain their certification guarantees) can be implemented in two ways: (1) sequential or (2) batched. The first setting, which performs the 10,000 inferences sequentially, introduces no memory overhead (still 5.3 MB), but takes 5.41 seconds per image, which is an 8,300$\\times$ runtime overhead; this setting assumes one has a device that cannot support large batch processing, as is common in real-time, edge settings. If one does have the resources to batch process the 10,000 inferences in parallel, the runtime is 0.0153 seconds per image, which is still slower than our inference by a factor of 23$\\times$ – however, this setting incurs a significant memory overhead of 3.4 GB GPU memory. Various tradeoffs between memory/time are possible by combining the scenarios (1) and (2), but a large resource cost (whether it be runtime or memory) is inevitable with randomized smoothing.
> > >
> > > We have clarified in the Related Work section that, at inference time, deterministically certified networks are significantly faster *or* more memory-efficient. Also, this distinction between randomized smoothing and deterministic certification is *not specific to our geometric setting*, and a discussion of the two approaches' merits can also be seen here: https://openreview.net/forum?id=SJxSDxrKDr&noteId=HJlVRSuADr. Thank you for your insightful comments that have helped us improve the paper.

---

> > > > ### Comment · Reviewer_Z6j5 · 2022-11-18
> > > > **Thanks for your response**
> > > >
> > > > Thank you - it was very helpful to have the numbers here. (And yes, your point on randomized smoothing vs. determinstic certification is well taken -- I asked because I wanted to get a better sense for how to fairly compare the two approaches in the general setting).

---

> ### Author Response · Authors · 2022-11-14
> **Response to Reviewer Z6j5 (Part 1 of 2)**
>
> Thank you for taking the time to review our paper and for providing us with valuable, constructive feedback and insightful questions – we greatly appreciate it! We have revised the paper according to your advice (with all changes in blue), and our responses to your questions are as follows.
>
> **A minor concern I have is around the hyperparameter $\\nu$. The schedule of values for these hyperparameter are provided in the Appendix, but it's not clear to me how the work arrives at these values in the first place.**
>
> We agree with the reviewer that this detail should be addressed in the paper; thank you for pointing this out. We have added a detailed procedure to Appendix D.4 (which we refer the reader to in Section 5, “Hyperparameters”) on how we obtained the appropriate $\\nu$ values. In short, we first compute training images’ maximum pixel interval width under perturbation (to gauge the amount of over-approximation) and tune $\\nu$ such that these widths are not too wide (i.e., comparable to typical $\\ell_\\infty$-ball sizes); for each experimental setting, this computation can be performed in *under a minute* and *before any training*. After this procedure, similar to standard hyperparameter tuning, we fine-tune $\\nu$ by performing a train-validation split, training and certifying the network, and tuning $\\nu$ for accuracy and robustness. We only had to do this step a few times (and thus it does not add significant overhead), as the calculation of the pixel widths already provided a good heuristic for network performance. The complete details are in Appendix D.4.
>
> **Q: How do we define the output for geometric transformations (any rotation, any translation, shrinking, any shearing) that move some parts of the image out of the frame and leave some areas of the frame with no corresponding source pixel in the image?**
>
> Consistent with the state-of-the-art deterministic geometric verifier DeepG (Balunovic et al., 2019) that we compare our verification results to, we use zero padding where we set areas of the frame with no corresponding source pixel to zero. We have made this clear in Section 4.2 (bottom of page 5).
>
> Conceptually, our algorithm is agnostic to the particular padding mode: to handle other padding techniques (e.g., replicating the border of the in-frame part of the image), our core algorithm remains *completely unchanged* – all that is needed is a simple preprocessing step that pads the input image according to the desired padding strategy (before being fed into our algorithm), and a postprocessing step that crops the output back to the original dimensions. We have added a discussion of this in Appendix C.2.
>
> **Q: Are interpolated transformations bijective? Otherwise, how do you handle this? ... When rotating this output matrix by another 45 degrees, which of the original matrices should you obtain?**
>
> Interpolated transformations are not bijective: for the reviewer’s example, both inputs rotated by 45 degrees indeed result in the same output of $\\begin{bmatrix} 0.5 & 0.5 \\\\ 0.5 & 0.5 \\end{bmatrix}$ (assuming replication padding). When further rotating this output by another 45 degrees, the output is neither of the original matrices: it is still $\\begin{bmatrix} 0.5 & 0.5 \\\\ 0.5 & 0.5 \\end{bmatrix}$. On the other hand, if we were to rotate the input $\\begin{bmatrix} 1 & 0 \\\\ 0 & 1 \\end{bmatrix}$ directly by 90 degrees, we would obtain $\\begin{bmatrix} 0 & 1 \\\\ 1 & 0 \\end{bmatrix}$; similarly, rotating $\\begin{bmatrix} 0 & 1 \\\\ 1 & 0 \\end{bmatrix}$ by 90 degrees results in $\\begin{bmatrix} 1 & 0 \\\\ 0 & 1 \\end{bmatrix}$. These differences are caused by interpolation – rotation and other interpolated transformations do not compose: $\\mathrm{Rotate}(X, 90^\\circ) \\neq \\mathrm{Rotate}(\\mathrm{Rotate}(X, 45^\\circ), 45^\\circ)$ in general for an image $X$. This is because for the left-hand side, the interpolation function will be called only once, while for the right-hand side, the interpolation function will be called twice. This phenomenon is not specific to our interval setting, but to interpolated transformations in general.

---

> > ### Comment · Reviewer_Z6j5 · 2022-11-15
> > **Thanks for addressing my concerns + one follow up question**
> >
> > This was a good paper before the changes and continues to be a good paper after. I liked the author's approach of showing their edits in blue - that really helps reviewers track the changes that have been made. (I'm learning a lot as a reviewer in this cycle!)
> >
> > **hyperparameter tuning**
> >
> > Thanks for the detailed section in the appendix. I hope it helps other researchers build on your work (rather than spending a lot of time figuring out how to get the magic numbers, as they might have had before!)
> >
> > **padding**
> >
> > That makes sense.
> >
> > **Are interpolated transformations bijective?**
> >
> > I see. If I'm understanding correctly, the order in which we apply the transformations (e.g. $T_x, T_y$) is important to get right as it affects the final output (and possibly the provable robustness)? [1]
> >
> > [1] As you noted, this is just a property of the definition of geometric transformations in general?

---

> > > ### Author Response · Authors · 2022-11-17
> > > **Thank you + answering your follow-up question**
> > >
> > > Thank you for your positive comments! To answer your question, yes: the order in which geometric transformations are applied affects the image output; this is a property of their definition in general and *not a phenomenon specific to our work*. There are two reasons:
> > >
> > > 1. Rounding effects due to multiple interpolations (as shown in your example above). Note that this situation does *not* arise in our work, because we consider the whole composition of transformations together and only interpolate *once* afterward. We have revised the text above Eq. 1 in Section 3.1 to clarify this.
> > > 2. The affine transformations defining the inverse mapping of $(x, y)$ coordinates (Eqs. 10-13) are not commutative in general, hence the order in which they are applied matters. (However, in the special case given by your example ($T_x(h)$, $T_y(v)$), the order of these two transformations does not matter because either way the final inverse coordinate mapping is $(x, y) \\mapsto (x-h, y-v)$.) For each benchmark, we apply affine transformations in the order they are listed (i.e., same order as DeepG).

---

### Decision · Program_Chairs · 2023-01-20

**Decision:**

Accept: notable-top-25%

**Justification For Why Not Higher Score:**

While the technique developed is clearly effective, it is not yet demonstrated to be effective on SOTA models/datasets.

**Justification For Why Not Lower Score:**

The problem of certified robustness to geometric transformations is well motivated, and the authors develop a rigorous solution backed up by several clever ideas for speeding up verification, enabling efficient implementation.

**Metareview: Summary, Strengths And Weaknesses:**

The authors develop a technique to obtain certified robustness against geometric transformations (like rotations) of inputs to image classifiers, developing an accelerated verified that can run on a GPU.

Strengths:
1. Paper is well written and experiments are convincing.
2. The technique has efficient GPU implementations and clear psuedocode.

Weaknesses:
1. Details of the bound propagation procedure were unclear in initial versions as pointed out by some reviewers. However, these were adequately addressed in the rebuttal phase.

Hence, I recommend acceptance and encourage the authors to incorporate the (minor) comments from the reviewers in the final version.

**Note From Pc:**

if the above contains the word "oral" or "spotlight" please see: "oral" presentation means -> notable-top-5% and "spotlight" means -> notable-top-25%. As stated in our emails, we are disassociating presentation type from AC recommendations

**Summary Of Ac-Reviewer Meeting:**

No meeting